# Estimation of Treatment Effects Under Nonstationarity
# via the Truncated Policy Gradient Estimator

**Ramesh Johari** [1]    **Tianyi Peng** [2]    **Wenqian Xing** [1]

## Abstract

Randomized experiments (or A/B tests) are widely used to evaluate interventions in dynamic systems such as recommendation platforms, marketplaces, and digital health. In these settings, interventions affect both current and future system states, so estimating the global average treatment effect (GATE) requires accounting for temporal dynamics, which is especially challenging in the presence of *nonstationarity*; existing approaches suffer from high bias, high variance, or both. In this paper, we address this challenge via the novel Truncated Policy Gradient (TPG) estimator, which replaces instantaneous outcomes with short-horizon outcome trajectories. The estimator admits a *policy gradient* interpretation: it is a truncation of the first-order approximation to the GATE, yielding provable reductions in bias and variance in nonstationary Markovian settings. We further establish a central limit theorem for the TPG estimator and develop a consistent variance estimator that remains valid under nonstationarity with single-trajectory data. We validate our theory with two real-world case studies. The results show that relative to existing approaches, a well-calibrated TPG estimator can achieve a favorable balance between bias and variance in nonstationary settings, highlighting the value of the policy-gradient perspective for designing effective estimators under complex dynamics.

## 1. Introduction

Randomized controlled experiments ("A/B tests") are essential tools for evaluating the impact of interventions across dynamic, technology-driven environments such as recommendation systems, online marketplaces, and digital health platforms. Typically, these interventions do not merely influence immediate outcomes but also propagate effects over subsequent system states. This leads to a breakdown of the standard assumption of unit-level independence, a phenomenon known as a *carryover effect* (Bojinov et al., 2023; Hu & Wager, 2022) or *temporal interference* (Glynn et al., 2020). In such settings, it is well established that applying a naive estimator (e.g., the difference-in-means or Horvitz-Thompson estimators (Horvitz & Thompson, 1952)) can result in substantial bias as it ignores these intertemporal dependencies. For instance, in a digital health platform, randomization of incoming patients to treatment and control affects the workload of clinical providers, which in turn impacts the care experience of subsequent patients. Similarly, in a ride-sharing platform, user assignments to treatment and control affect the availability of drivers: a booking decision by one user influences the system state—e.g., driver distribution or availability—which in turn affects the experience and behavior of subsequent users.

Crucially, these real-world systems are also typically *nonstationary*: their dynamics can vary in a time-dependent manner, driven by factors such as evolving user preferences, time-dependent driver dynamics, and external shocks (e.g., seasonal trends or viral events). In the presence of nonstationarity, the problem of experimental design and estimation becomes even more challenging – a major issue for practitioners. In this paper, our central contribution is a simple and practical approach that leverages a policy-gradient perspective to yield an estimator that yields a favorable balance of bias and variance relative to existing approaches, in settings with both carryover effects and nonstationarity.

One way to model such dynamic, nonstationary environments is via a *Markovian* system, represented by a tuple sequence $(X_t, Z_t, Y_t)$ for a finite time horizon $t = 1, \ldots, T$. Here, $X_t$ denotes the system state at time $t$ (e.g., the number and locations of available drivers), and $Z_t \in \{0, 1\}$ indicates whether the user arriving at time $t$ is assigned to the control or treatment group. The variable $Y_t$ captures the reward resulting from the user's behavior after receiving the assignment (e.g., $Y_t = 1$ if the user books a trip). Conditioned on the state $X_t$ and action $Z_t$, the system evolves

---
[1]Department of Management Science and Engineering, Stanford University [2]Columbia Business School, Columbia University. Correspondence to: Wenqian Xing <wxing@stanford.edu>.

*Proceedings of the 43$^{rd}$ International Conference on Machine Learning*, Seoul, South Korea. PMLR 306, 2026. Copyright 2026 by the author(s).

according to:

$$X_{t+1} \sim P_{Z_t}^t(X_t, \cdot),$$

where $P_0^t$ and $P_1^t$ denote the (unknown) transition kernels under control and treatment at time $t$. The superscript $t$ reflects the system's nonstationarity.

Platforms typically want to compare two policies: the *treatment policy*, which assigns $Z_t = 1$ for all $t$, and the *control policy*, which assigns $Z_t = 0$ for all $t$. The estimand of interest is the expected average reward under each policy over a fixed finite time horizon, commonly referred to as the *global average treatment effect* (GATE). Since it is infeasible to observe both policies simultaneously, data are typically collected under an exploratory policy (the *experiment*)—for instance, a Bernoulli randomization policy that assigns each $Z_t$ independently to $0$ or $1$ with equal probability, as in a typical A/B test setting.

This yields a challenging causal inference problem: *estimation of the GATE from a single trajectory of the nonstationary system under the experiment.* A variety of approaches, of differing levels of sophistication, exist for this problem:

- **Naive estimator.** A naive approach is to simply ignore the interference and take the difference in average outcomes between treatment and control groups (e.g., a difference-in means or Horvitz-Thompson estimator (Horvitz & Thompson, 1952)). Though straightforward, the bias of such an estimator can be as large as the treatment effect itself (Glynn et al., 2020; Farias et al., 2022; Hu & Wager, 2022), which motivates the line of research aimed at addressing interference in the first place.

- **Stationary OPE.** Recent research has explored more sophisticated techniques inspired by reinforcement learning, including off-policy evaluation and the recently proposed difference-in-Q's (DQ) estimator (Farias et al., 2022; 2023; Uehara et al., 2022; Shi et al., 2023). However, most of these methods assume a *stationary* environment. When applied to our setting, they tend to exhibit high bias and substantial variance (see Sections 4 and 7), as they rely heavily on assumptions such as the stationary observability of system states or the availability of multiple independent trajectories—none of which hold in our setting. Moreover, existing analyses for stationary environments do not extend directly to nonstationary systems.

- **More complex designs.** An orthogonal line of research investigates more elaborate experimental designs, such as switchback experiments (Hu & Wager, 2022), to address nonstationarity. However, these designs can be challenging to implement in practice and often require careful tuning of parameters such as interval length. This motivates our investigation into whether improvements are possible within the more commonly utilized Bernoulli randomized design in A/B testing.

Perhaps surprisingly, we find that a simple modification of the difference-in-means (DM)[1] estimator yields a promising low-bias, low-variance alternative. Specifically, rather than comparing *instantaneous* outcomes (as in the DM estimator), we compare *k-step accumulated rewards*, with $k$ fixed (and chosen to be small relative to the experiment horizon).

To motivate our approach, we show that this difference of $k$-step sums can be viewed as an approximate *policy gradient*: namely, the gradient of the cumulative reward with respect to the probability of assignment to treatment. We thus refer to our estimator as the *Truncated Policy Gradient* (TPG) estimator (see Section 3 for the formal definition). By a Taylor approximation argument, this policy gradient approximates the GATE. For statistical inference, we establish a nonstationary central limit theorem for the TPG estimator and propose a consistent variance estimator that remains valid under nonstationarity with mild assumptions.

We find that it offers several appealing properties:

- **Theoretical.** We show that the TPG estimator achieves substantial variance reduction, at the cost of a controlled mixing bias, under mild assumptions on the environment's mixing time. Our analysis is based on a first-order policy-gradient formulation tailored to nonstationary environments, which may be of independent interest to the reinforcement learning community. We also establish a CLT and develop a *practical*, *consistent* variance estimator for single-trajectory nonstationary data.

- **Practical.** The estimator is simple to implement:
  1. It is based entirely on *observed rewards* and does not require prior knowledge of the state space;
  2. It can be computed from a *single sample path*, without requiring multiple i.i.d. trajectories;
  3. It allows for *post-experiment calibration*. The variance estimator we develop can be used to build confidence intervals and guides the choice of truncation level $k$. Specifically, we propose a heuristic by which the truncation size $k$ can be evaluated post-experiment, enabling practitioners to adaptively select the truncation level (see, e.g., Appendix B).

- **Empirical.** In Section 7, we empirically evaluate the TPG estimator using two realistic simulations: a queueing model based on nonstationary patient arrivals to a hospital emergency department, and a large-scale ride-sharing system based on NYC taxi data. Both settings reflect real-world service system dynamics. In this highly nonstationary environment, we compare its performance to standard OPE methods and a recent switchback-based

---

[1]Strictly speaking, this is a naive Horvitz–Thompson estimator, but we use the more common language of "difference-in-means," understanding that when $T$ is large, the gap between the two naive estimators is minimal.

approach (Hu & Wager, 2022). With appropriate calibration, our estimator consistently exhibits substantially lower bias and variance, demonstrating its robustness and practical utility in real-world settings.

**Related work.** Experimental design has received growing attention in the era of technology-driven environments, particularly in *networked systems* (Eckles et al., 2017; Peng et al., 2025; Aronow & Samii, 2017; Viviano et al., 2023) and *online marketplaces* (Johari et al., 2022; Wu et al., 2024a; Johari et al., 2024; Bojinov et al., 2023; Wager & Xu, 2021). Recent research has increasingly adopted *Markovian environments* to model the structural dynamics of treatment and control in experiments with temporal dependencies (Glynn et al., 2020; Hu & Wager, 2022; Farias et al., 2022; 2023; Li et al., 2023). Among these, the difference-in-Q's (DQ) estimator (Farias et al., 2022; 2023) has demonstrated a favorable bias-variance tradeoff under stationary Markovian settings. Experiments and inference in *nonstationary environments* have also received increasing attention. Prior work has examined nonstationary time series models (Simchi-Levi et al., 2023; Wu et al., 2024b) and nonstationary Markovian environments with geometric mixing-time properties (Hu & Wager, 2022). Our model closely aligns with the setting considered in (Hu & Wager, 2022). Additionally, off-policy evaluation (OPE) methods have been widely studied in both fully observed and partially observed MDP settings (Hu & Wager, 2023; Uehara et al., 2022; 2023; Shi et al., 2023; Thomas & Brunskill, 2016; Su et al., 2020). However, a key limitation is that these methods typically rely on multiple independent trajectories and are difficult to adapt to a single trajectory in a nonstationary Markovian environment due to the curse of horizon (Uehara et al., 2022).

## 2. Preliminaries

In this section, we present the formal framework: Section 2.1 introduces the nonstationary Markovian model, and Section 2.2 defines the global average treatment effect (GATE) and the Bernoulli randomized design.

### 2.1. A Dynamic Model of Treatment

**Notation.** We write $f(x) = O(g(x))$ if $\exists C > 0$ s.t. $|f(x)| \leq C|g(x)|$ for all large $x$, and $f(x) = o(g(x))$ if $|f(x)|/|g(x)| \to 0$; for random $\{X_n\}$ and deterministic $\{a_n\}$, $X_n = O_p(a_n)$ if $X_n/a_n$ is bounded in probability and $X_n = o_p(a_n)$ if $X_n/a_n \to 0$ in probability. We define $[T] := \{1, \ldots, T\}$ and $\mathrm{Law}(Y)$ for the distribution of $Y$.

**Time horizon.** We consider a discrete, finite time horizon indexed by $t = 1, \ldots, T$.

**State and action.** We consider a finite state space $\mathcal{X}$ and a binary action space $\mathcal{Z} = \{0, 1\}$.

**Transition kernel.** For $x \in \mathcal{X}$ and $z \in \mathcal{Z}$, we let $P_z^t(x, x')$ denote the transition probability from $x$ to $x'$ if action $z$ is chosen at time $t$, i.e., $\mathbb{P}(X_{t+1} = x'|X_t = x, Z_t = z) = P_z^t(x, x')$.

We assume the following *mixing time* property on the transition kernels. A similar uniform mixing-time bound was assumed by (Farias et al., 2022) and (Hu & Wager, 2022) in the context of switchback experiments in a Markovian setting. More broadly, mixing time assumptions are common in the literature on bandits and reinforcement learning (Van Roy, 1998; Even-Dar et al., 2004; Hu & Wager, 2023), as well as in the study of general Markov processes (Doukhan, 1994).

**Assumption 2.1** (Mixing time). There exists a constant $\gamma \in (0, 1)$, such that, for any time $t \in [T]$, any action $z \in \mathcal{Z}$, and any pair of distributions $f$ and $f'$ over the state space $\mathcal{X}$, the following holds

$$\left\| f'P_z^t - fP_z^t \right\|_{\mathrm{TV}} \leq \gamma \left\| f' - f \right\|_{\mathrm{TV}},$$

where $fP_z^t$ denotes the distribution on $\mathcal{X}$ defined by $(fP_z^t)(x') := \sum_{x \in \mathcal{X}} f(x)P_z^t(x, x')$, and $\| \cdot \|_{\mathrm{TV}}$ denotes the total variation distance.

*Remark* 2.2. Similar mixing time assumptions are standard in the contextual bandits and reinforcement learning literature (Van Roy, 1998; Even-Dar et al., 2004; Hu & Wager, 2022; 2023), and have been used in both stationary and time-varying MDPs (Hu & Wager, 2022). Moreover, real-world service systems such as ride-sharing exhibit approximate mixing, with the influence of a single treatment dissipating over time under continual arrivals (Curry et al., 2019).

**Markov process.** We denote the distribution of the initial state $X_1$ by $\rho$. For all time steps, the state transitions satisfy the Markov property with time-dependent transition kernels.

**Rewards.** At each time $t$, a real-valued random reward $Y_t$ is generated with mean $r(x, z)$ when action $z$ is taken in state $x$. We assume $Y_t$ is uniformly bounded by a constant $M$, and, conditional on $(X_t, Z_t)$, the reward noise is independent of both the past and the transition dynamics.

### 2.2. Estimand and Experimental Design

In this section, we describe our estimand as well as a Bernoulli randomized experimental design to compare treatment and control.

**Markov policies.** We consider a family of Markov policies $\{\pi_\theta : \theta \in (0, 1)\}$. Under $\pi_\theta$, action $z = 1$ is chosen with probability $\theta$ (and $z = 0$ with probability $1 - \theta$), independently of state and past randomness. The extremes $\pi_1$ and $\pi_0$ correspond to the *treatment* and *control* policies, respectively, while policies with $0 < \theta < 1$ assign actions i.i.d. as Bernoulli($\theta$), which we refer to as *Bernoulli randomization*.

For such a policy, the transition kernel at time $t$ is $P_\theta^t = \theta P_1^t + (1 - \theta)P_0^t$, and the expected reward in state $x$ is $r_\theta(x) = \theta r(x, 1) + (1 - \theta)r(x, 0)$. We denote by $\mathcal{L}_\theta^t$ the law of $Y_t$ under $\pi_\theta$, i.e.,

$$\mathcal{L}_\theta^t = \text{Law}\left(Y_t \mid X_1 \sim \rho, \{Z_u\}_{u=1}^t \sim \text{i.i.d. Bern}(\theta)\right).$$

**Estimand.** Let $X_1 \sim \rho$ be the initial state. The treatment effect at time $t$ is $\tau_t := \mathbb{E}_{\mathcal{L}_1^t}[Y_t] - \mathbb{E}_{\mathcal{L}_0^t}[Y_t]$, and the global average treatment effect (GATE) over horizon $[T]$ is

$$\tau := \frac{1}{T}\sum_{t=1}^T \tau_t.$$

**Data generation and estimator.** The experiment starts from an initial state $X_1 \sim \rho$, evolves under a uniform Bernoulli randomization policy $\pi_{1/2}$ over the time horizon $[T]$ (for simplicity, we assume $\theta = 1/2$; the analysis extends straightforwardly to any $\theta \in (0, 1)$), and generates a single data trajectory $\{(X_t, Z_t, Y_t) : 1 \le t \le T\}$. An *estimator* computed at time $T$ is any random variable adapted to $\sigma((X_t, Z_t, Y_t) : 1 \le t \le T)$.

## 3. Estimation Under Nonstationarity

A naive approach to estimating $\tau$ under a Bernoulli randomization policy $\pi_{1/2}$ is to use the following difference-in-means (DM) estimator:

$$\hat{\tau}_0 := \frac{1}{T}\sum_{u=1}^T \left(\frac{\mathbf{1}\{Z_u = 1\}}{1/2} - \frac{\mathbf{1}\{Z_u = 0\}}{1/2}\right)Y_u. \quad (1)$$

The DM estimator is unbiased and exhibits low variance when the observations are i.i.d. (Horvitz & Thompson, 1952; Robins et al., 1994). However, in dynamic (e.g., Markovian) environments, the DM estimator has been shown to incur substantial bias due to temporal *interference* across observations (Glynn et al., 2020; Farias et al., 2022; Hu & Wager, 2022). Consequently, a variety of estimators have been developed to correct for temporal interference in dynamic settings, including the recently proposed difference-in-Q's (DQ) estimators (Farias et al., 2022; 2023) and estimators based on off-policy evaluation (OPE) techniques (Shi et al., 2023; Uehara et al., 2022). Despite these advances, most bias-corrected estimators still rely on stationarity assumptions and can fail in nonstationary environments, where bias correction is inherently more challenging.

**Truncated Policy Gradient (TPG).** To address the challenge of estimation under nonstationarity, we propose an estimator that naturally extends the naive DM estimator and, perhaps surprisingly, mitigates temporal interference bias even in nonstationary Markovian environments. Our approach is straightforward to implement: instead of adding

up only the immediate outcome $Y_t$ at each time $t$ as in (1), we sum the subsequent outcomes over a window of size $k$. Formally, for each $0 \le k \le T$, we define the estimator:

$$\hat{\tau}_k := \frac{1}{T}\sum_{u=1}^T \left(\frac{\mathbf{1}\{Z_u = 1\}}{1/2} - \frac{\mathbf{1}\{Z_u = 0\}}{1/2}\right)\sum_{t=u}^{\min(u+k,T)} Y_t. \quad (2)$$

Note that when $k = 0$, the estimator $\hat{\tau}_k$ reduces to the DM estimator, which completely ignores the nonstationary Markovian dynamics of the environment. In contrast, when $k \ge 1$, the estimator $\hat{\tau}_k$ captures the Markovian dynamics by incorporating short-term future outcomes into the estimation. We refer to $\hat{\tau}_k$ as the *truncated policy gradient estimator* with truncation size $k$; later in Section 5, we show that $\mathbb{E}[\hat{\tau}_k]$ represents the gradient of a hypothetical truncated policy.

*Remark* 3.1. The TPG estimator $\hat{\tau}_k$ does not require access to the states, offering a practical advantage over estimators that depend on full state observability (e.g., OPE estimators (Uehara et al., 2022)). Moreover, when there is no interference, $\hat{\tau}_k$ provides an unbiased estimation of $\tau$ under any truncation size $k$ (reduces to DM as $k = 0$).

The main results of this paper establish bias and variance bounds for the TPG estimator $\hat{\tau}_k$ in nonstationary Markovian environments, as formalized in Theorem 3.2.

*Theorem* 3.2 (Bias and variance bounds for the TPG estimator). *Under Assumption 2.1, the bias of the TPG estimator $\hat{\tau}_k$ with respect to the GATE $\tau$, for any truncation size $0 \le k \le T$, is bounded by*

$$\left|\mathbb{E}[\hat{\tau}_k] - \tau\right| = O\left(\frac{1 - \gamma^k}{(1 - \gamma)^2}\delta^2 M + \frac{\gamma^k}{1 - \gamma}\delta M\right), \quad (3)$$

*where $\delta$ is the* kernel deviation, *i.e., the maximum total variance distance between the transition kernels across all states and times: $\delta := \sup_{1 \le t \le T}\sup_{x \in \mathcal{X}}\|P_1^t(x, \cdot) - P_0^t(x, \cdot)\|_{\text{TV}}$. The variance of $\hat{\tau}_k$ is bounded by*

$$\text{Var}(\hat{\tau}_k) = O\left(\frac{(k+1)^3 M^2}{T} + \frac{\gamma(k+1)^2 M^2}{T(1 - \gamma)}\right). \quad (4)$$

Theorem 3.2 shows that the bias and variance of $\hat{\tau}_k$ is jointly governed by the truncation size $k$, the mixing rate $\gamma$, and the kernel deviation $\delta$. In particular, since the DM estimator corresponds to $k = 0$, the theorem demonstrates that the bias reduction benefit of the TPG estimator with $k > 0$ depends on the relationship between $\delta$, $\gamma$, and $k$. By rearranging, we see that the TPG estimator with $k \ge 1$ admits a lower bias bound than the DM estimator whenever $\delta < 1 - \gamma$. That is, the TPG estimator with $k \ge 1$ effectively debiases the DM estimator when the treatment and control kernels are closer, i.e., when $\delta$ is smaller, or when the process mixes faster, i.e., when $\gamma$ is smaller.

$$
\begin{array}{c}
\text{(PG Estimator)} \qquad \text{(Policy Gradient)} \\
\mathbb{E}[\hat{\tau}_T] \quad = \quad \nabla J(1/2) \underset{\text{(Taylor error)}}{\approx} J(1) - J(0) \quad = \quad \tau \text{ (GATE)} \\
\Big\updownarrow \text{ (Mixing bias)} \\
\text{(Taylor error)} \\
\mathbb{E}[\hat{\tau}_k] \quad = \quad \nabla J_k(1/2) \approx J_k(1) - J_k(0) \\
\text{(TPG Estimator)} \qquad \text{(Truncated Policy Gradient)}
\end{array}
$$

*Figure 1.* Connections between the TPG, PG estimators, policy gradients, and the GATE.

We derive this theorem over the next two sections. We start in Section 4 by first introducing a broader class of (untruncated) policy gradient (PG) estimators in the nonstationary environment described in Section 2. The policy gradient can be viewed as a first-order Taylor approximation to the GATE (i.e., the difference in policy value for $\pi_1$ and $\pi_0$). In this way, we obtain a bias bound for the untruncated PG estimator in the nonstationary setting that (1) is dominated by the Taylor error; and (2) unfortunately, scales quadratically in $T$ due to nonstationarity.

Our analysis of the TPG estimator is then carried out in a parallel manner, and shows this poor bias behavior can be mitigated. We show in Section 5 that the TPG estimator can be interpreted as an estimator of a *hypothetical truncated policy gradient*, that estimates the difference between two policies: first playing $\pi_{1/2}$, and then playing $\pi_1$ or $\pi_0$ respectively in the most recent $k$ time periods. Because we approximate this difference using the hypothetical truncated policy gradient, a Taylor approximation error is again incurred, similar to the untruncated version; this is the first term in the bias bound (3) in the theorem, which we refer to as the *Taylor error*. The second term in the bias bound arises by bounding the gap between the hypothetical truncated policy difference and the GATE, taking advantage of the mixing time assumption Assumption 2.1; thus, we refer to the second term as the *mixing bias*. Crucially, we observe that due to truncation, the estimator $\hat{\tau}_k$ has a bias that scales quadratically in $k$, rather than $T$ as for the untruncated PG estimator. Figure 1 shows the connections among TPG, PG estimators, policy gradients, and the GATE.

## 4. The Policy Gradient Estimator Under Nonstationarity

As a starting point in our analysis, we consider the untruncated policy gradient (PG) estimator, i.e., $k = T$. We show how this estimator can be naturally interpreted as an estimation of the *policy gradient* (Marbach & Tsitsiklis, 2001; Sutton, 2018) for the Bernoulli randomization policy, and use this interpretation to quantify its bias via a Taylor approximation argument. We begin with several preliminary definitions, following standard notation from the reinforce-

ment learning literature.

**Value function.** Define the *value function* under the Bernoulli randomized policy $\pi_\theta$ as the average outcome over the horizon $[T]$, i.e., in our setting with nonstationary Markovian environments,

$$
J(\theta) := \frac{1}{T} \sum_{t=1}^{T} \mathbb{E}_{\mathcal{L}_\theta^t}[Y_t]. \tag{5}
$$

Note with this definition that the GATE $\tau$ is equal to $J(1) - J(0)$. We then define the $Q$-value of taking action $z$ at time $t$ under the experimental policy $\pi_{1/2}$ as the expected cumulative reward from time $t$ to $T$. Formally, we write:

$$
Q_{1/2}^t(z) := \sum_{u=t}^{T} \mathbb{E}_{\mathcal{L}_{1/2}^t}[Y_u \mid Z_t = z]. \tag{6}
$$

Note that this definition of the $Q$-value in nonstationarity is *state-independent*. Unlike the standard $Q$-value in infinite-horizon stationary MDPs that depends on the state-action pair, our definition reflects the expected value from taking action $z$ at time $t$, conditioned on the initial state $X_1 \sim \rho$.

*Proposition 4.1.* The policy gradient of $J(\theta)$ in (5) evaluated at the uniform randomized policy $\pi_{1/2}$ exists and is given by the average difference in $Q$-values defined in (6) over the horizon, i.e.,

$$
\nabla J(1/2) = \frac{1}{T} \sum_{t=1}^{T} \left( Q_{1/2}^t(1) - Q_{1/2}^t(0) \right). \tag{7}
$$

Proposition 4.1 shows that the policy gradient $\nabla J(1/2)$ corresponds to the average $Q$-value difference over the horizon $[T]$.[2] In fact, the difference-in-Q's (DQ) estimator proposed in (Farias et al., 2022) can be interpreted as a policy gradient in stationary average-reward MDPs (see Appendix C for a detailed discussion).

Along these lines, the key step in our analysis of the untruncated PG estimator $\hat{\tau}_T$ is to view it as an estimator for

---

[2]In stationary MDPs, $J(\theta)$ is known to be smooth under both average-reward and discounted-reward formulations (Sutton, 2018); our theorem shows it remains smooth for finite-horizon non-stationary MDPs.

the policy gradient $\nabla J(1/2)$ in the nonstationary setting. Applying a Taylor expansion around $J(1/2)$, the bias of $\hat{\tau}_T$ relative to $\tau$ decomposes as

$$
\mathbb{E}[\hat{\tau}_T] - \tau = \underbrace{\left( \mathbb{E}[\hat{\tau}_T] - \nabla J(1/2) \right)}_{\text{gradient estimation bias}}
$$
$$
+ \underbrace{\sum_{n=2}^{\infty} \frac{1}{n!} \nabla^n J(1/2) \left[ (-\tfrac{1}{2})^n - (\tfrac{1}{2})^n \right]}_{\text{Taylor remainder}}, \quad (8)
$$

where the Taylor error corresponds to the higher-order bias in approximating $\tau$ using the policy gradient $\nabla J(1/2)$. Now observe by elementary calculation that the untruncated PG estimator is *unbiased* for the policy gradient $\nabla J(1/2)$, i.e., $\mathbb{E}[\hat{\tau}_T] = \nabla J(1/2)$. Therefore, the entire bias is due to the Taylor error. The following proposition shows that both the bias and the variance of the untruncated PG estimator scale *quadratically* in the full horizon $T$, without requiring Assumption 2.1.

*Proposition 4.2. Without imposing Assumption 2.1, the bias of $\hat{\tau}_T$ with respect to the GATE $\tau$ is bounded by*

$$
|\mathbb{E}[\hat{\tau}_T] - \tau| = O\big(T^2 \delta^2 M\big). \quad (9)
$$

*In fact, the preceding bound applies for any unbiased estimator $\hat{\tau}$ of $\nabla J(1/2) = J'(1/2)$. Moreover, the variance of $\hat{\tau}_T$ is bounded as $\mathrm{Var}(\hat{\tau}_T) = O\big(T^2 M^2\big)$.*

Proposition 4.2 can be viewed as the untruncated counterpart of Theorem 3.2 in the absence of Assumption 2.1. Together, the two results clarify the distinct roles of mixing and truncation. The mixing condition yields a uniform control of the bias, but does not by itself reduce the variance of the untruncated estimator. Truncation is introduced precisely to reduce this variance, improving the order from $O(T^2 M^2)$ for the untruncated estimator to the corresponding truncated order, at the cost of an additional bias term that decays geometrically in the truncation length. Thus, the advantage of the TPG estimator lies in its favorable bias–variance trade-off: it achieves substantial variance reduction while incurring only a controlled additional mixing bias. The later numerical results in Section 7 demonstrate this trade-off clearly.

## 5. Estimation via the Truncated Policy Gradient Estimator

As shown in Proposition 4.2, the performance of the untruncated PG estimator is poor over larger time horizons $T$; due to the nonstationarity of the environment both bias and variance grow quadratically. In this section, we show that the TPG estimator exhibits more favorable performance in this regime. In particular, we carry out a parallel analysis of

the TPG estimator (2) with general $k$, and show how we can leverage the mixing time (Assumption 2.1) to obtain more favorable bounds on bias and variance.

We start by introducing the hypothetical truncated policy gradient in nonstationary MDPs and establish its connection to the TPG estimator $\hat{\tau}_k$.

**Hypothetical truncated policy.** Let $\mathcal{L}_\theta^{t,k}$ denote the distribution of the outcome $Y_t$ when policy $\pi_{1/2}$ is followed for the first $t - (k+1)$ states (if $t > k+1$), and policy $\pi_\theta$ is followed for the remaining $k$ states (i.e., a hypothetical truncated policy), conditioned on the initial state distribution $X_1 \sim \rho$, we have:

$$
\mathcal{L}_\theta^{t,k} = \mathrm{Law} \left( Y_t \, \middle| \, \begin{array}{l} X_1 \sim \rho, \\ \{Z_u\}_{u=1}^{t-k-1} \sim \text{i.i.d. Bern}(1/2), \\ \{Z_u\}_{u=t-k}^{t} \sim \text{i.i.d. Bern}(\theta) \end{array} \right).
$$
$$
(10)
$$

The *truncated policy value function* $J_k(\theta)$ is defined as

$$
J_k(\theta) := \frac{1}{T} \sum_{t=1}^{T} \mathbb{E}_{\mathcal{L}_\theta^{t,k}}[Y_t]. \quad (11)
$$

We refer to the gradient of $J_k(\theta)$ as the *truncated policy gradient*. For truncation size $0 \leq k \leq T$, define the *truncated Q-value* as

$$
Q_{1/2}^{t,k}(z) := \sum_{u=t}^{\min(t+k,T)} \mathbb{E}_{\mathcal{L}_{1/2}^{t}} \left[ Y_u \mid Z_t = z \right]. \quad (12)
$$

*Proposition 5.1. The truncated policy gradient $\nabla J_k(\theta)$ evaluated at the uniform randomized policy $\pi_{1/2}$ exists and is given by the average difference in truncated Q-values in (12) over the horizon, i.e.,*

$$
\nabla J_k(1/2) = \frac{1}{T} \sum_{t=1}^{T} \left( Q_{1/2}^{t,k}(1) - Q_{1/2}^{t,k}(0) \right).
$$

With the TPG estimator $\hat{\tau}_k$ defined in (2), we have

$$
\mathbb{E}[\hat{\tau}_k] = \frac{1}{T} \sum_{t=1}^{T} \left( Q_{1/2}^{t,k}(1) - Q_{1/2}^{t,k}(0) \right) = \nabla J_k(1/2).
$$
$$
(13)
$$

This indicates that the TPG estimator $\hat{\tau}_k$ can be viewed as an estimation of $\tau$ via an unbiased estimation of the truncated policy gradient $\nabla J_k(1/2)$. Accordingly, the bias of $\hat{\tau}_k$ relative to $\tau$ decomposes as

$$
\mathbb{E}\left[\hat{\tau}_k\right] - \tau = \underbrace{\nabla J_k(1/2) - (J_k(1) - J_k(0))}_{\text{Taylor error w.r.t. } J_k(1/2)}
$$
$$
+ \underbrace{(J_k(1) - J(1)) + (J(0) - J_k(0))}_{\text{mixing bias}},
$$

which includes a Taylor error between $\nabla J_k(1/2)$ and $J_k(1) - J_k(0)$, and an additional mixing bias between the original value function $J(\cdot)$ and its truncated counterpart $J_k(\cdot)$. We note that the Taylor error is of order $O(k^2)$, as the treatment probability affects only the most recent $k$ states rather than the full trajectory (cf. the $O(T^2)$ bias of $\hat{\tau}$).

Figure 1 illustrates the connections among the GATE, the policy gradients $\nabla J(1/2)$ and $\nabla J_k(1/2)$, and the corresponding PG estimator $\hat{\tau}_T$ and TPG estimator $\hat{\tau}_k$. The bias of the TPG estimator $\hat{\tau}_k$ involves an additional mixing bias but reduces the Taylor error. The variance of $\hat{\tau}_k$ can be bounded by applying the mixing-time property in Assumption 2.1 within a strong $\alpha$-mixing framework (Rosenblatt, 1956). Unlike the variance bound for $\hat{\tau}$ that grows with $T$, the variance of $\hat{\tau}_k$ remains bounded in the asymptotic regime as $T \to \infty$ for any fixed truncation size $k$.

**Choice of size $k$.** Post-calibration methods such as Lepski's method (Lepski et al., 1997; Su et al., 2020) can select $k$ in a data-driven way using the variance estimates in Section 6, but should carefully handle the non-monotone bias bound. As a practical guideline, we suggest starting with the standard DM estimator (i.e., $k = 0$) and gradually increasing $k$ until the variance grows more rapidly than the change in the estimation. Our numerical results show that, in most cases, increasing $k$ corrects the bias of the DM estimator and moves the estimate closer to the ground-truth GATE, thereby revealing the bias direction. A practical heuristic $k$-selection algorithm is provided in Appendix B.

## 6. Nonstationary CLT and Variance Estimator

In this section, we establish the asymptotic normality of the TPG estimator $\hat{\tau}_k$ for any finite truncation size $k$ and present a consistent nonstationary estimator of its asymptotic variance $\sigma_k^2$. Define $w_t^{\text{IPW}} := 2(2Z_t - 1)$, $B_t := \sum_{i=\max\{1, t-k\}}^{t} w_i^{\text{IPW}} Y_t$, and $\bar{B} := T^{-1} \sum_{t=1}^{T} B_t$. For any finite $k$, we consider the following assumptions to regularize the nonstationarity:

*Assumption 6.1 (Asymptotic variance).* We assume that the following limit exists and is finite and positive:

$$\sigma_k^2 := \lim_{T \to \infty} \frac{1}{T} \text{Var}\left(\sum_{t=1}^{T} B_t\right) \in (0, \infty).$$

*Assumption 6.2 (Cesàro-$L^2$ mean stability).* Define $\mu_t := \mathbb{E}[B_t]$ and the averaged mean $\bar{\mu}_T := T^{-1} \sum_{t=1}^{T} \mu_t$. Assume that the Cesàro-$L^2$ deviation satisfies

$$\Delta_T^2 := \frac{1}{T} \sum_{t=1}^{T} (\mu_t - \bar{\mu}_T)^2 = o(T^{-2/3}).$$

Assumptions 6.1 and 6.2 impose mild regularity on the system's nonstationarity; i.e., Assumption 6.1 ensures that

*Table 1.* Empirical coverage of 95% confidence interval (CI) using the nonstationary HAC variance estimator in Case Study I. The choice $k = 5$ is selected by the $k$-selection algorithm in Appendix B, which yields near-nominal 95% coverage.

| $k$ | 2 | 3 | 4 | **5** | 6 | 7 |
|---|---|---|---|---|---|---|
| Coverage (%) | 2.0 | 44.2 | 93.2 | **94.6** | 85.6 | 78.8 |

the relevant empirical variances converge in large samples, while Assumption 6.2 controls the mean drift of $B_t$ via a Cesàro-$L^2$ condition. Under these conditions, Theorem 6.3 establishes the CLT for the TPG estimator and presents a *nonstationary heteroskedasticity- and autocorrelation-consistent* (HAC) variance estimator.

*Theorem 6.3.* Fix a finite truncation size $k \geq 0$. Under Assumptions 2.1, 6.1, and 6.2, define $\hat{V}_t := B_t - \bar{B}$,

$$\hat{\Gamma}_\ell := \frac{1}{T} \sum_{t=1}^{T-\ell} \hat{V}_t \hat{V}_{t+\ell}, \quad w_\ell^{\text{NW}} := 1 - \frac{\ell}{L_T + 1},$$

where $L_T \to \infty$ with $L_T = O(T^{1/3})$. We construct the nonstationary HAC variance estimator as

$$\hat{\Omega}_T := \hat{\Gamma}_0 + 2 \sum_{\ell=1}^{L_T} w_\ell^{\text{NW}} \hat{\Gamma}_\ell.$$

Then, as $T \to \infty$,

$$\sqrt{T} (\hat{\tau}_k - \mathbb{E}[\hat{\tau}_k]) \Rightarrow \mathcal{N}(0, \sigma_k^2), \quad \hat{\Omega}_T \xrightarrow{p} \sigma_k^2.$$

*Proof sketch.* The proof of Theorem 6.3 has two parts: (i) establish a CLT under a nonstationary MDP, and (ii) show the consistency of the proposed nonstationary HAC variance estimator. A key challenge is that the traditional HAC estimator (Hansen, 1992) is infeasible with single-trajectory nonstationary data, as $\mathbb{E}[B_t]$ cannot be consistently estimated. We overcome this by working with time-averaged blocks $B_t - \bar{B}$ and leveraging Cesàro-$L^2$ mean stability, which controls the time-variation of $\mathbb{E}[B_t]$ and ensures that the proposed nonstationary HAC estimator converges to $\sigma_k^2$.

## 7. Numerical Results

In this section, we evaluate the TPG estimator in three settings: (i) a queueing simulator calibrated to real-world nonstationary patient arrivals (adapted from (Li et al., 2023)); (ii) a large-scale NYC ride-sharing simulator using real trip data (adapted from (Peng et al., 2025)); and (iii) a synthetic two-state nonstationary Markov decision process (see Appendix G.1). All code and experiment details are publicly available.[3]

---

[3] https://github.com/wenqian-xing/TPG-Estimator

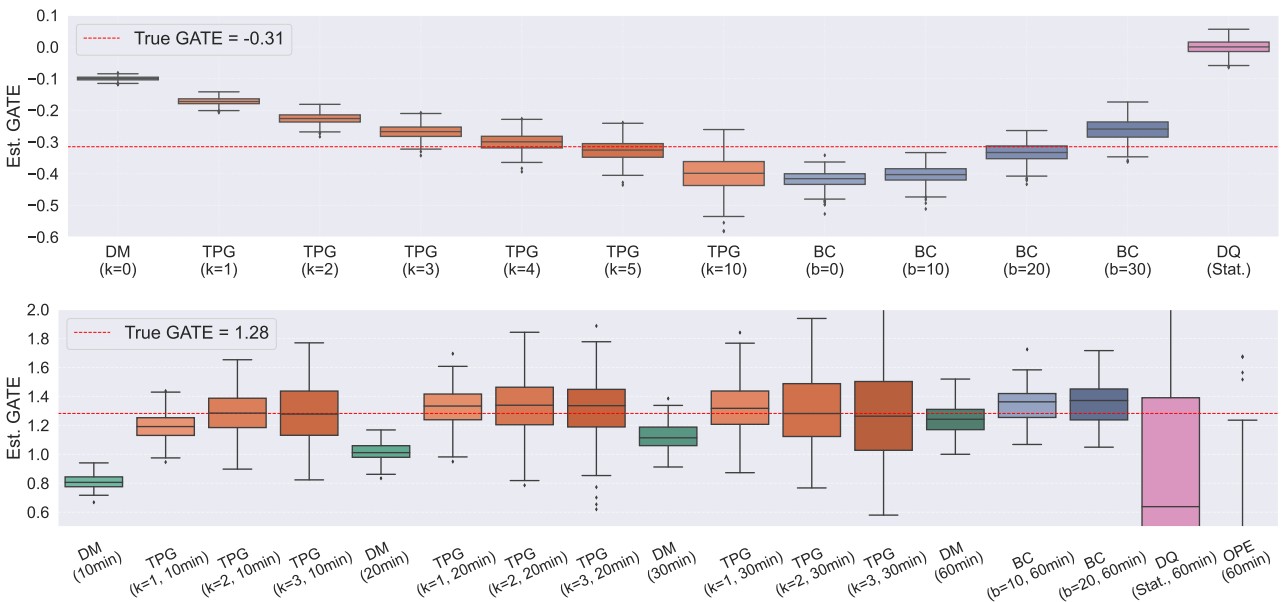

*Figure 2.* Estimation performance under two realistic settings: real-world nonstationary patient arrivals (top) and NYC ride-sharing simulation with real data from September 2024 (bottom).

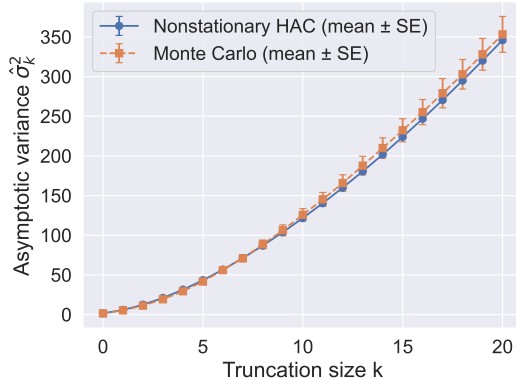

*Figure 3.* Variance estimates for the TPG estimator $\hat{\tau}_k$ in a realistic patient queueing system in Case Study I.

**Case study I: Queueing simulator based on real-world nonstationary patient arrivals.** To test the real-world effectiveness of our estimator, we implement a queueing simulator with nonstationary arrivals following (Li et al., 2023). The simulator is a birth–death chain with inhomogeneous, state-dependent Poisson arrivals, estimated from emergency department data in the SEEStat Home Hospital database (SEE-Center, 2013) (varying by day and time). Customers are less likely to join when the state is higher (i.e., the emergency department is busier), while the service rate remains constant. We use a discrete-time version with a constant proportional shift in arrival rates between treatment and control. The simulator runs for 40,320 steps (four weeks of 1-minute steps) across 500 experiment trajectories, with switchback intervals of 1 hour. Full details are in Appendix G.2.

We compare the TPG estimator with the stationary DQ and LSTD-OPE (Farias et al., 2022), and with the bias-corrected (BC) estimator (with burn-in) under the Bernoulli *switchback design* (Hu & Wager, 2022). See Figure 2. LSTD-OPE yields a mean of $-2.39$, outside the plotted range. Both DM and stationary DQ exhibit substantial bias. By contrast, the TPG estimator with $k = 4$ or $5$ (chosen by Appendix B) closely approximates the GATE. The BC estimator with a 20-minute burn-in also performs well, but it depends on a more complex switchback design that typically requires prior knowledge to choose the interval.

Table 1 and Figure 3 further validate the nonstationary HAC variance estimator in this case study via Monte Carlo simulation: it closely matches the simulated variance, and the resulting 95% CI attains near-nominal coverage at $k = 5$.

**Case study II: NYC ride-sharing simulator.** We further adapt a large-scale NYC ride-sharing simulator using real-world data from (Peng et al., 2025). Rider arrivals and destination locations are generated based on historical records provided by New York City's Taxi and Limousine Commission (TLC, 2024), restricted to trips within Manhattan. In this setting, we evaluate the platform's pricing policy, where the fare offered to a rider is computed as a fixed rate per unit of time multiplied by the estimated trip duration from pickup to destination. The treatment policy applies a higher rate per unit of time than the control policy. Upon receiving a price quote, the rider can either accept the trip—triggering a dispatch of the nearest available vehicle and generating a reward equal to the trip fare—or reject it, resulting in no dispatch and zero reward. The simulator runs for 500,000 ar-

rivals (142 hours), across 100 i.i.d. trajectories. Full details are in Appendix G.3.

We use a Bernoulli switchback design, assigning each interval (e.g., 10–60 minutes) independently to treatment or control with probability $1/2$. The system state is the average number of available drivers. Figure 2 compares estimators. The DM estimator is highly sensitive to interval length, i.e., short intervals (10–30 minutes) yield large bias, while 60 minutes performs well. In contrast, the TPG estimator consistently reduces bias across intervals, with only modest variance increase, making it a robust post-processing tool. Among state-of-the-art estimators, BC offers little improvement over DM under 1-hour intervals and slightly overestimates with small burn-in, while stationary DQ and OPE fail to learn stable $Q$-values, yielding high bias and variance in estimation.

In addition, we can observe the bias–variance trade-off described after Proposition 4.2, i.e., increasing the truncation size $k$ leads to a significant increase in variance, whereas the mixing bias induced by choosing a smaller $k$ appears to be modest. In particular, in the ride-sharing case study, the TPG estimator with $k = 2$ or $k = 3$ already produces estimates that are very close to the GATE.

## 8. Conclusion

In this paper, we introduced the TPG estimator for the GATE in nonstationary Markovian settings, which operates by differencing $k$-step reward aggregates. We developed a nonstationary HAC variance estimator and established a corresponding CLT for the TPG estimator, enabling both data-driven selection of the truncation size $k$ and valid statistical inference. Finally, through theoretical analysis and two realistic case studies, we showed that under mild mixing-time conditions, TPG achieves low bias and variance in single-trajectory, nonstationary Markov environments, without requiring prior knowledge of the state space or multiple independent trajectories, and that the proposed variance estimator provides a consistent asymptotic variance estimation.

## Acknowledgments

The authors are especially grateful to Andrew Zheng for his contributions to the ride-sharing simulator, to Shuangning Li for generously sharing details of the nonstationary patient-arrival queueing simulator, and to Stefan Wager for insightful discussions on the policy gradient theorem. We also gratefully acknowledge the SEE Research Team at the Technion-Israel Institute of Technology for providing the queueing data used in our simulations. This work was supported in part by Stanford Data Science, the Stanford Human-Centered AI Institute, and the National Science Foundation under Grant 2205084.

## Impact Statement

This paper develops a methodology for estimating treatment effects in dynamic, nonstationary systems from a single randomized experimental trajectory. More reliable estimation under temporal carryover and time-varying dynamics can improve decision-making in applications such as digital health, online marketplaces, and recommendation or pricing systems by reducing the risk of deploying interventions whose apparent benefits are artifacts of interference or nonstationarity. The societal considerations of this work are those common to experimentation and policy evaluation used to guide product and operational changes.

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

# Appendix Contents

Appendix A introduces the notation and technical lemmas used throughout the proofs. Appendix B presents a heuristic algorithm for selecting the truncation size $k$. Appendix C discusses the stationary average-reward MDP setting and the DQ estimator. Appendices D, E, and F provide detailed proofs of the main theorems, propositions, and lemmas, respectively. Appendix G reports additional experimental results and describes the experimental setup in detail.

# A. Notation and Technical Lemmas

In this section, we present the key notation along with several technical results and lemmas used throughout the proofs.

## A.1. Time-Varying Transition Kernels

We denote by $P_\theta^{a \to b}$ the transition kernel induced by the Bernoulli-randomized policy $\pi_\theta$ from time $a$ to time $b$ for $1 \le a \le b \le T$, defined as

$$P_\theta^{a \to b} := \begin{cases} \prod_{u=a}^{b-1} P_\theta^u, & \text{if } a < b, \\ I, & \text{if } a = b, \end{cases}$$

where $I$ denotes the identity matrix. Additionally, note that $P_\theta^{a \to a+1} = P_\theta^a$ for all $a \in [T]$.

## A.2. Time-Varying Kernel Deviations

For each time step $t \in [T]$, we define the difference in transition kernels as $\Delta^t := P_1^t - P_0^t$, where $P_1^t$ and $P_0^t$ are the transition kernels under the always-treated and always-controlled policies, respectively. We quantify the maximum deviation across time and states using the worst-case total variation norm:

$$\delta^t := \sup_{x \in \mathcal{X}} \left\| P_1^t(x, \cdot) - P_0^t(x, \cdot) \right\|_{\text{TV}}, \quad \delta := \sup_{1 \le t \le T} \sup_{x \in \mathcal{X}} \left\| P_1^t(x, \cdot) - P_0^t(x, \cdot) \right\|_{\text{TV}}.$$

Additionally, we define the deviations from the uniform randomization policy $\pi_{1/2}$ at each time $t$ as $\Delta_+^t := P_1^t - P_{1/2}^t$ and $\Delta_-^t := P_0^t - P_{1/2}^t$, where $P_{1/2}^t = (P_1^t + P_0^t)/2$. It follows directly that

$$\sup_{1 \le t \le T} \sup_{x \in \mathcal{X}} \left\| \Delta_+^t(x, \cdot) \right\|_{\text{TV}} = \sup_{1 \le t \le T} \sup_{x \in \mathcal{X}} \left\| \Delta_-^t(x, \cdot) \right\|_{\text{TV}} = \sup_{1 \le t \le T} \sup_{x \in \mathcal{X}} \frac{1}{2} \left\| \Delta^t(x, \cdot) \right\|_{\text{TV}} \le \delta.$$

## A.3. Derivative of the Transition Kernels

Recall that under the Bernoulli-randomized policy $\pi_\theta$, the transition kernel and expected reward at time $t$ are given by

$$P_\theta^t = (1 - \theta) P_0^t + \theta P_1^t, \quad r_\theta = (1 - \theta) r_0 + \theta r_1.$$

Differentiating with respect to the treatment probability $\theta$, we have

$$\frac{d}{d\theta} P_\theta^t = P_1^t - P_0^t = \Delta^t, \quad \frac{d}{d\theta} r_\theta = r_1 - r_0.$$

By applying the product rule for matrix-valued functions for all $1 \le a < b \le T$, we have

$$\frac{d}{d\theta} P_\theta^{a \to b} = \frac{d}{d\theta} \prod_{u=a}^{b-1} P_\theta^u = \sum_{u=a}^{b-1} \left( \prod_{v=a}^{u-1} P_\theta^v \right) \left( \frac{d}{d\theta} P_\theta^u \right) \left( \prod_{v=u+1}^{b-1} P_\theta^v \right) = \sum_{u=a}^{b-1} P_\theta^{a \to u} \Delta^u P_\theta^{(u+1) \to b},$$

and $\frac{d}{d\theta} P_\theta^{a \to a} = 0$ for all $a \in [T]$.

## A.4. Strong Mixing Property

We introduce the strong ($\alpha$) mixing property (Rosenblatt, 1956) and relevant lemmas used in our variance analysis.

*Lemma* A.1 (Lemma 3 in (Doukhan, 1994)). *Let $X$ and $Y$ be measurable random variables with respect to sigma-algebras $\mathcal{U}$ and $\mathcal{V}$, respectively. Denote $\|X\|_\infty = \inf\{M \geq 0 : \mathbb{P}(|X| > M) = 0\}$. Then,*

$$|\mathrm{Cov}(X, Y)| \leq 4\,\alpha(\mathcal{U}, \mathcal{V})\|X\|_\infty\|Y\|_\infty,$$

*where*

$$\alpha(\mathcal{U}, \mathcal{V}) = \sup_{A \in \mathcal{U}, B \in \mathcal{V}} \left| \mathbb{P}(A \cap B) - \mathbb{P}(A)\mathbb{P}(B) \right|.$$

*Lemma* A.2. *Under Assumption 2.1, the process $\{(X_t, Z_t, Y_t)\}_{t=1}^T$ is $\alpha$-mixing (Rosenblatt, 1956) under Bernoulli randomization policy $\pi_\theta$. Specifically, for any $1 \leq a \leq b \leq T$, define the sigma-algebra*

$$\mathcal{F}_a^b := \sigma\big((X_t, Z_t, Y_t) \colon a \leq t \leq b\big),$$

*which represents the information generated by the states, actions, and outcomes from time $a$ to $b$. Then, for every lag $h \geq 1$, the $\alpha$-mixing coefficient is*

$$\alpha(h) = \sup_{1 \leq s \leq T-h} \sup_{\substack{A \in \mathcal{F}_1^s \\ B \in \mathcal{F}_{s+h}^T}} \left| \mathbb{P}(A \cap B) - \mathbb{P}(A)\mathbb{P}(B) \right| \leq \gamma^{h-1}.$$

*Proof of Lemma A.2.* Fix any lag $h \geq 1$ and any split point $1 \leq s \leq T - h$. Let

$$\mathcal{F}_1^s = \sigma\left((X_1, Z_1, Y_1), \ldots, (X_s, Z_s, Y_s)\right), \ \mathcal{F}_{s+h}^T = \sigma\left((X_{s+h}, Z_{s+h}, Y_{s+h}), \ldots, (X_T, Z_T, Y_T)\right).$$

By definition, we have

$$\alpha(h) = \sup_{1 \leq s \leq T-h} \sup_{\substack{A \in \mathcal{F}_1^s \\ B \in \mathcal{F}_{s+h}^T}} \left| \mathbb{P}(A \cap B) - \mathbb{P}(A)\mathbb{P}(B) \right|.$$

We show that $\left| \mathbb{P}(A \cap B) - \mathbb{P}(A)\mathbb{P}(B) \right| \leq \gamma^{h-1}$ for any $A \in \mathcal{F}_1^s$ and $B \in \mathcal{F}_{s+h}^T$.

First, we observe

$$\mathbb{P}(A \cap B) = \mathbb{E}\left[\mathbf{1}\{A\}\mathbf{1}\{B\}\right] = \mathbb{E}\left[\mathbb{E}\left[\mathbf{1}\{A\}\mathbf{1}\{B\} \mid \mathcal{F}_1^s\right]\right] = \mathbb{E}\left[\mathbf{1}\{A\}\mathbb{P}(B \mid \mathcal{F}_1^s)\right],$$

and

$$\mathbb{P}(A)\mathbb{P}(B) = \mathbb{E}\left[\mathbf{1}\{A\}\mathbb{P}(B)\right].$$

Then, we have

$$|\mathbb{P}(A \cap B) - \mathbb{P}(A)\mathbb{P}(B)| \leq \mathbb{E}\left[\mathbf{1}\{A\}|(\mathbb{P}(B \mid \mathcal{F}_1^s) - \mathbb{P}(B))|\right]$$

$$\overset{\dagger}{\leq} \mathbb{P}(A) \sup_{\substack{x,x' \in \mathcal{X} \\ z,z' \in \mathcal{Z}}} |\mathbb{P}(B \mid X_s = x, Z_s = z) - \mathbb{P}(B \mid X_s = x', Z_s = z')|$$

$$\overset{\dagger}{=} \mathbb{P}(A) \sup_{\substack{x,x' \in \mathcal{X} \\ z,z' \in \mathcal{Z}}} \left| \sum_{y \in \mathcal{X}} \mathbb{P}(B \mid X_{s+h} = y)\Big(\mathbb{P}(X_{s+h} = y \mid X_s = x, Z_s = z) \right.$$

$$\left. - \mathbb{P}(X_{s+h} = y \mid X_s = x', Z_s = z')\Big) \right|$$

$$\overset{\ddagger}{\leq} \mathbb{P}(A) \sup_{\substack{x,x' \in \mathcal{X} \\ z,z' \in \mathcal{Z}}} \left\| (e_x P_z^s - e_{x'} P_{z'}^s) \prod_{t=s+1}^{s+h-1} P_\pi^t \right\|_{\mathrm{TV}}$$

$$\overset{*}{\leq} \mathbb{P}(A)\gamma^{h-1}$$

$$\leq \gamma^{h-1},$$

where $e_x$ denotes the probability distribution vector with a one in the $x$th position and zero elsewhere. Here, $(\dagger)$ follows from the Markov property and the independence of the reward, $(\ddagger)$ follows from the dual representation of total variation, and $(*)$ follows from the mixing-time property (Assumption 2.1). $\qquad\square$

*Corollary* A.3. *Under Lemma A.2 and Bernoulli randomization policy $\pi_\theta$, define*

$$\Gamma_t := 2\left(2Z_t - 1\right) \sum_{u=t}^{\min(t+k,T)} Y_u \in \sigma\big((X_t, Z_t, Y_t), \ldots, (X_{t+k}, Z_{t+k}, Y_{t+k})\big) = \mathcal{F}_t^{t+k}.$$

*Then the process $\{\Gamma_t\}_{t=1}^T$ is $\alpha$-mixing with the mixing coefficient for any lag $h \geq 1$ is*

$$
\begin{aligned}
\alpha_\Gamma(h) &= \sup_{\substack{1 \leq s \leq T-h}} \sup_{\substack{A \in \sigma(\Gamma_{1:s}) \\ B \in \sigma(\Gamma_{s+h:T})}} \big|\mathbb{P}(A \cap B) - \mathbb{P}(A)\mathbb{P}(B)\big| \\
&\leq \sup_{\substack{1 \leq s \leq T-h}} \sup_{\substack{A \in \mathcal{F}_1^{s+k} \\ B \in \mathcal{F}_{s+h}^T}} \big|\mathbb{P}(A \cap B) - \mathbb{P}(A)\mathbb{P}(B)\big| \\
&\leq \begin{cases} 1, & 1 \leq h \leq k+1, \\ \alpha(h-k), & h \geq k+2, \end{cases} \\
&\leq \begin{cases} 1, & 1 \leq h \leq k+1, \\ \gamma^{h-k-1}, & h \geq k+2, \end{cases} \\
&= \gamma^{\max\{0, h-k-1\}}.
\end{aligned}
$$

## B. Heuristic Truncation Size $k$-Selection Algorithm

---
**Algorithm 1** Heuristic $k$-Selection via Stability

---
**Input:** Candidate set $\mathcal{K} = \{0, 1, 2, \ldots, K_{\max}\}$; stability threshold $\alpha \geq 0$
**Output:** Selected $k^\star$
$k^\star \leftarrow \varnothing$
$(\hat{\tau}_0, \hat{\sigma}_0) \leftarrow$ TPG (or DM) and nonstationary HAC estimator under truncation size 0
**for** $k = 1$ **to** $K_{\max}$ **do**
  $(\hat{\tau}_k, \hat{\sigma}_k) \leftarrow$ TPG and nonstationary HAC estimator under truncation size $k$
  **if** $|\hat{\tau}_k - \hat{\tau}_{k-1}| \leq \alpha\,\hat{\sigma}_k$ **then**
    $k^\star \leftarrow k$
    **break**
  **end if**
**end for**
**if** $k^\star = \varnothing$ **then**
  $k^\star \leftarrow 0$                                          # Use the naive DM estimator
**end if**
**return** $k^\star$

---

Algorithm 1 is a practical heuristic in the spirit of Lepski's method (Lepski et al., 1997). It selects the largest truncation size $k$ for which the increase in variance does not exceed the change in the estimated treatment effect. The resulting $k^\star$ balances bias reduction with variance control, ensuring the estimator is effectively debiased without being dominated by the quadratically increasing variance.

**Choice of $\alpha$.** We adopt $\alpha = 1$ as the default, corresponding to an approximate $70\%$ confidence level. This threshold controls the stopping point in the bias–variance trade-off: larger values lead to later termination, allowing for lower bias at the cost of higher variance. Table 2 reports a robustness check across conventional confidence levels for Case Study I in Section 7. The selected median $k$ and the resulting RMSE remain stable across these choices, suggesting that the procedure is not sensitive to the precise calibration of $\alpha$.

*Table 2.* Sensitivity of post-$k$ selection to the hyperparameter $\alpha$.

| $\alpha$ | Confidence level | Median $k$ | RMSE |
|---|---|---|---|
| 1.036 | 70% | 5 | 0.043 |
| 1.282 | 80% | 4 | 0.039 |
| 1.645 | 90% | 4 | 0.040 |
| 1.960 | 95% | 3 | 0.050 |

*Notes.* The threshold $\alpha$ is the standard normal critical value associated with each two-sided confidence level. RMSE is computed against the true GATE $\tau$ in Case Study I across simulation replications.

## C. DQ Estimator (Farias et al., 2022) in a Stationary Average-Reward MDP

In (Farias et al., 2022), a *difference-in-Q's* (DQ) estimator is proposed to estimate the average treatment effect (ATE) in a stationary, average-reward Markovian setting. This approach was later extended to finite-horizon and discounted-reward settings in (Farias et al., 2023). Recall that in an average-reward MDP, for a state $x$ and action $z$, the $Q$-value $Q(x, z|\theta)$ is defined as the *excess* reward over the long-run average reward obtained starting from state $x$ with initial action $z$, under policy $\pi_\theta$. The DQ estimator assumes access to an unbiased estimator of the $Q$-function $Q(\cdot|\theta)$ and computes the difference in steady-state $Q$-values between the treatment and control actions. The DQ estimator has been shown to achieve a favorable bias-variance trade-off in stationary MDPs, especially when $Q$-values can be efficiently estimated without bias (e.g., with LSTD-$\lambda$ (Bradtke & Barto, 1996)). By the policy gradient theorem for stationary average-reward MDPs (Sutton, 2018), when $J(\cdot)$ denotes the steady-state value function, its gradient $\nabla J(\theta)$ can be written in terms of the policy gradient, which in our case (since actions are binary) can be expressed as a difference in $Q$-values. Thus we can view the DQ estimator as a form of policy-gradient estimator in the stationary setting. We demonstrate this in detail below.

**Policy gradient in an average-reward MDP.** In an average-reward MDP, the $Q$-value represents the cumulative excess reward of a state-action pair $(x, z)$ relative to the long-run average reward under a Bernoulli-randomized policy $\pi_\theta$. Specifically, it is defined as

$$Q(x, z|\theta) := \lim_{T \to \infty} \mathbb{E}_{\pi_\theta} \left[ \sum_{i=1}^{T} (Y_i - \lambda_\theta) \mid X_1 = x, Z_1 = z \right],$$

where $\mathbb{E}_{\pi_\theta}$ denotes the expectation under policy $\pi_\theta$, and $\lambda_\theta$ denotes the long-run average return under policy $\pi_\theta$, i.e., $\lambda_\theta := \lim_{T \to \infty} \frac{1}{T} \mathbb{E}_{\pi_\theta} \left[ \sum_{i=1}^{T} Y_i \right]$. In this setting, the $Q$-value depends on the state and action, but no longer on time.

Under the ergodicity assumption, let $\rho_\theta$ denote the stationary distribution of the states induced by policy $\pi_\theta$. The steady-state value function $J(\theta)$ is then defined as

$$J(\theta) := \sum_{x \in \mathcal{X}} \rho_\theta(x) \sum_{z \in \mathcal{Z}} \pi_\theta(z|x) r(x, z).$$

Then, the average treatment effect (ATE) in (Farias et al., 2022) is defined as $\tau := J(1) - J(0)$. By the policy gradient theorem for average-reward MDPs (see Page 333 in (Sutton, 2018)), we have

$$\begin{aligned} \nabla J(\theta) &= \sum_{x \in \mathcal{X}} \rho_\theta(x) \sum_{z \in \mathcal{Z}} \nabla \pi_\theta(z|x) Q(x, z|\theta) \\ &= \sum_{x \in \mathcal{X}} \rho_\theta(x) \left( Q(x, 1|\theta) - Q(x, 0|\theta) \right), \end{aligned}$$

which corresponds exactly to the DQ estimator proposed in (Farias et al., 2022) when $\theta = 1/2$.

## D. Proof of Theorems

### D.1. Proof of Theorem 3.2

*Proof of Theorem 3.2.* The key idea here is to estimate the global treatment effect at each time using the transition kernel under the experimental policy $\pi_{1/2}$, and to show that this estimate corresponds to a truncated policy gradient (or truncated difference in $Q$-values by Proposition 5.1). We first prove the bias bound, followed by the variance bound.

**Bias bound.** Recall that $\hat{\tau}_k$ is unbiased for $\nabla J_k(1/2)$, so it suffices to bound

$$|\mathbb{E}[\hat{\tau}_k] - \tau| = |\nabla J_k(1/2) - \tau|.$$

For each $t \in [T]$, define

$$s_t := \max\{1, t - k\}, \qquad \ell_t := t - s_t = \min\{k, t - 1\}.$$

We first record two operator bounds.

Since

$$P^u_{1/2} = \frac{1}{2}P^u_1 + \frac{1}{2}P^u_0,$$

Assumption 2.1 implies that $P^u_{1/2}$ inherits the same contraction coefficient $\gamma$: for any two probability distributions $f, f'$ on $\mathcal{X}$,

$$\|f'P^u_{1/2} - fP^u_{1/2}\|_{\mathrm{TV}} \le \frac{1}{2}\|f'P^u_1 - fP^u_1\|_{\mathrm{TV}} + \frac{1}{2}\|f'P^u_0 - fP^u_0\|_{\mathrm{TV}} \le \gamma\|f' - f\|_{\mathrm{TV}}.$$

By Jordan decomposition, the same bound extends to all signed measures $\mu$ with $\mu(\mathcal{X}) = 0$; hence for all $1 \le a \le b \le T$,

$$\|\mu P^{a \to b}_z\|_{\mathrm{TV}} \le \gamma^{b-a}\|\mu\|_{\mathrm{TV}}, \qquad z \in \{0, 1, 1/2\}. \tag{14}$$

Next, by the definition of $\delta$ and again by Jordan decomposition, for every signed measure $\mu$,

$$\|\mu\Delta^u_+\|_{\mathrm{TV}} \le \delta\|\mu\|_{\mathrm{TV}}, \qquad \|\mu\Delta^u_-\|_{\mathrm{TV}} \le \delta\|\mu\|_{\mathrm{TV}}. \tag{15}$$

Fix $1 \le a < b \le T$, and let $\mu^\top$ be any probability row vector. Using the telescoping identity for the difference of two matrix products,

$$\mu^\top P^{a \to b}_1 - \mu^\top P^{a \to b}_{1/2} = \sum_{u=a}^{b-1} \mu^\top P^{a \to u}_1 \Delta^u_+ P^{(u+1) \to b}_{1/2}. \tag{16}$$

Subtracting the first-order term

$$\sum_{u=a}^{b-1} \mu^\top P^{a \to u}_{1/2} \Delta^u_+ P^{(u+1) \to b}_{1/2},$$

we obtain the exact remainder

$$\begin{aligned}
R^{(+)}_{a,b}(\mu) &:= \mu^\top P^{a \to b}_1 - \mu^\top P^{a \to b}_{1/2} - \sum_{u=a}^{b-1} \mu^\top P^{a \to u}_{1/2} \Delta^u_+ P^{(u+1) \to b}_{1/2} \\
&= \sum_{u=a}^{b-1} \mu^\top \left(P^{a \to u}_1 - P^{a \to u}_{1/2}\right) \Delta^u_+ P^{(u+1) \to b}_{1/2} \\
&= \sum_{a \le i < u \le b-1} \mu^\top P^{a \to i}_1 \Delta^i_+ P^{(i+1) \to u}_{1/2} \Delta^u_+ P^{(u+1) \to b}_{1/2},
\end{aligned} \tag{17}$$

where in the last step we applied (16) again to $P^{a \to u}_1 - P^{a \to u}_{1/2}$.

Now each summand in (17) is bounded, using (14) and (15), by

$$\begin{aligned}
&\left\|\mu^\top P^{a \to i}_1 \Delta^i_+ P^{(i+1) \to u}_{1/2} \Delta^u_+ P^{(u+1) \to b}_{1/2}\right\|_{\mathrm{TV}} \\
&\le \delta\gamma^{u-i-1} \delta\gamma^{b-u-1} = \delta^2\gamma^{u-i-1}\gamma^{b-u-1}.
\end{aligned}$$

Therefore,

$$\|R_{a,b}^{(+)}(\mu)\|_{\mathrm{TV}} \le \delta^2 \sum_{a \le i < u \le b-1} \gamma^{u-i-1} \gamma^{b-u-1}$$

$$= \delta^2 \sum_{m=0}^{b-a-2} (m+1)\gamma^m = \delta^2 \Psi_{b-a}(\gamma). \tag{18}$$

The same argument, with $P_0^u = P_{1/2}^u + \Delta_-^u$, yields the analogous identity and bound

$$\|R_{a,b}^{(-)}(\mu)\|_{\mathrm{TV}} \le \delta^2 \Psi_{b-a}(\gamma), \tag{19}$$

for the first-order perturbation remainder around $P_0$.

Fix $t \in [T]$. We decompose

$$\mathbb{E}_{\mathcal{L}_1^t}[Y_t] = \rho^\top P_1^{1 \to t} r_1$$

$$= \underbrace{\left(\rho^\top P_1^{1 \to s_t} - \rho^\top P_{1/2}^{1 \to s_t}\right) P_1^{s_t \to t} r_1}_{=:B_{1,t}} + \underbrace{\rho^\top P_{1/2}^{1 \to s_t} P_1^{s_t \to t} r_1}_{=:L_{1,t}}. \tag{20}$$

We first bound the tail term $B_{1,t}$. If $s_t = 1$, then $B_{1,t} = 0$. Otherwise $s_t = t - k$, and by the same telescoping argument used above,

$$\left\|\rho^\top P_1^{1 \to s_t} - \rho^\top P_{1/2}^{1 \to s_t}\right\|_{\mathrm{TV}} \le \sum_{u=1}^{s_t-1} \gamma^{s_t-u-1}\delta \le \frac{\delta}{1-\gamma}.$$

Since this signed measure has total mass zero, another application of (14) gives

$$\left\|\left(\rho^\top P_1^{1 \to s_t} - \rho^\top P_{1/2}^{1 \to s_t}\right) P_1^{s_t \to t}\right\|_{\mathrm{TV}} \le \gamma^k \frac{\delta}{1-\gamma}.$$

Using $\|r_1\|_\infty \le M$, we obtain the uniform bound

$$|B_{1,t}| \le \frac{\gamma^k}{1-\gamma} \delta M. \tag{21}$$

Next, apply (17)–(18) with $\mu^\top = \rho^\top P_{1/2}^{1 \to s_t}$, $a = s_t$, and $b = t$. Since $\rho^\top P_{1/2}^{1 \to s_t}$ is a probability row vector,

$$L_{1,t} = \rho^\top P_{1/2}^{1 \to t} r_1 + \sum_{u=s_t}^{t-1} \rho^\top P_{1/2}^{1 \to u} \Delta_+^u P_{1/2}^{(u+1) \to t} r_1 + \varepsilon_{1,t}, \tag{22}$$

where

$$|\varepsilon_{1,t}| \le \|R_{s_t,t}^{(+)}(\rho^\top P_{1/2}^{1 \to s_t})\|_{\mathrm{TV}} \|r_1\|_\infty \le \delta^2 M \, \Psi_{\ell_t}(\gamma) \le \delta^2 M \, \Psi_k(\gamma). \tag{23}$$

Combining (20), (21), (22), and (23), we obtain

$$\mathbb{E}_{\mathcal{L}_1^t}[Y_t] = \phi_1^t + O\left(\Psi_k(\gamma)\delta^2 M + \frac{\gamma^k}{1-\gamma}\delta M\right), \tag{24}$$

where

$$\phi_1^t := \rho^\top P_{1/2}^{1 \to t} r_1 + \sum_{u=\max(t-k,1)}^{t-1} \rho^\top P_{1/2}^{1 \to u} \Delta_+^u P_{1/2}^{(u+1) \to t} r_1.$$

Exactly the same argument, using (19), gives

$$\mathbb{E}_{\mathcal{L}_0^t}[Y_t] = \phi_0^t + O\left(\Psi_k(\gamma)\delta^2 M + \frac{\gamma^k}{1-\gamma}\delta M\right), \tag{25}$$

where

$$\phi_0^t := \rho^\top P_{1/2}^{1\to t} r_0 + \sum_{u=\max(t-k,1)}^{t-1} \rho^\top P_{1/2}^{1\to u} \Delta_-^u P_{1/2}^{(u+1)\to t} r_0.$$

*Step 4: identification with the truncated policy gradient.*

Exactly as in the original proof,

$$\phi_1^t - \phi_0^t = \sum_{u=\max(t-k,1)}^{t} \left( \mathbb{E}_{\mathcal{L}_{1/2}^t}[Y_t \mid Z_u = 1] - \mathbb{E}_{\mathcal{L}_{1/2}^t}[Y_t \mid Z_u = 0] \right).$$

Hence,

$$\frac{1}{T}\sum_{t=1}^{T}(\phi_1^t - \phi_0^t) = \frac{1}{T}\sum_{t=1}^{T}\sum_{u=\max(t-k,1)}^{t} \left( \mathbb{E}_{\mathcal{L}_{1/2}^t}[Y_t \mid Z_u = 1] - \mathbb{E}_{\mathcal{L}_{1/2}^t}[Y_t \mid Z_u = 0] \right)$$

$$= \frac{1}{T}\sum_{u=1}^{T}\sum_{t=u}^{\min(u+k,T)} \left( \mathbb{E}_{\mathcal{L}_{1/2}^t}[Y_t \mid Z_u = 1] - \mathbb{E}_{\mathcal{L}_{1/2}^t}[Y_t \mid Z_u = 0] \right)$$

$$= \nabla J_k(1/2),$$

where the last equality follows from Proposition 5.1.

Then, using (24) and (25),

$$|\mathbb{E}[\hat{\tau}_k] - \tau| = |\nabla J_k(1/2) - \tau|$$

$$= \left| \frac{1}{T}\sum_{t=1}^{T}(\phi_1^t - \phi_0^t) - \frac{1}{T}\sum_{t=1}^{T}\left( \mathbb{E}_{\mathcal{L}_1^t}[Y_t] - \mathbb{E}_{\mathcal{L}_0^t}[Y_t] \right) \right|$$

$$\leq \frac{1}{T}\sum_{t=1}^{T}\left( \left| \phi_1^t - \mathbb{E}_{\mathcal{L}_1^t}[Y_t] \right| + \left| \phi_0^t - \mathbb{E}_{\mathcal{L}_0^t}[Y_t] \right| \right)$$

$$= O\left( \Psi_k(\gamma)\delta^2 M + \frac{\gamma^k}{1-\gamma}\delta M \right).$$

Equivalently, for $k \geq 1$,

$$\Psi_k(\gamma) = \frac{1 - k\gamma^{k-1} + (k-1)\gamma^k}{(1-\gamma)^2} \leq \frac{1-\gamma^k}{(1-\gamma)^2}.$$

Therefore, the bias bound can be simplified as follows

$$|\mathbb{E}[\hat{\tau}_k] - \tau| = O\left( \frac{1-\gamma^k}{(1-\gamma)^2}\delta^2 M + \frac{\gamma^k}{1-\gamma}\delta M \right).$$

**Variance bound.** We now prove the variance bound.

For any truncation size $0 \leq k \leq T$, recall that $\hat{\tau}_k$ is given by

$$\hat{\tau}_k = \frac{1}{T}\sum_{u=1}^{T}\left( \frac{\mathbf{1}\{Z_u = 1\}}{1/2} - \frac{\mathbf{1}\{Z_u = 0\}}{1/2} \right) \sum_{t=u}^{\min(t+k,T)} Y_t.$$

For all $1 \leq t \leq T$, denote

$$\Gamma_t := \left( \frac{\mathbf{1}\{Z_t = 1\}}{0.5} - \frac{\mathbf{1}\{Z_t = 0\}}{0.5} \right) \sum_{u=t}^{\min(t+k,T)} Y_u = 2(2Z_t - 1) \sum_{u=t}^{\min(t+k,T)} Y_u.$$

Then, the variance of the TPG estimator $\hat{\tau}_k$ can be written as

$$\mathrm{Var}(\hat{\tau}_k) = \frac{1}{T^2} \left( \sum_{t=1}^{T} \mathrm{Var}(\Gamma_t) + 2 \sum_{1 \leq i < j \leq T} \mathrm{Cov}(\Gamma_i, \Gamma_j) \right).$$

We begin by bounding the variance term $\mathrm{Var}(\Gamma_t)$. For any time $t \in [T]$, we observe

$$\mathrm{Var}(\Gamma_t) = \mathbb{E}[\Gamma_t^2] - \mathbb{E}^2[\Gamma_t] \leq \mathbb{E}[\Gamma_t^2],$$

where $|\Gamma_t| \leq 2(k+1)M$.

Then,

$$\mathrm{Var}(\Gamma_t) \leq \mathbb{E}[\Gamma_t^2] \leq 4(k+1)^2 M^2.$$

Next, we bound the covariance term $\mathrm{Cov}(\Gamma_i, \Gamma_j)$ for $1 \leq i < j \leq T$. By Corollary A.3, the process $\{\Gamma_t\}$ is $\alpha$-mixing with mixing coefficients satisfying $\alpha_\Gamma(h) \leq \gamma^{\max(0, h-k-1)}$ for any lag $h \geq 1$. Then, applying the foundational covariance bound for $\alpha$-mixing sequences in Lemma A.1, we obtain

$$|\mathrm{Cov}(\Gamma_i, \Gamma_j)| \leq 4\alpha_\Gamma(j-i)\|\Gamma_i\|_\infty \|\Gamma_j\|_\infty \leq 16\gamma^{\max(0, j-i-k-1)}(k+1)^2 M^2.$$

Then, we have

$$\begin{aligned}
2 \sum_{1 \leq i < j \leq T} \mathrm{Cov}(\Gamma_i, \Gamma_j) &\leq 2 \sum_{1 \leq i < j \leq T} |\mathrm{Cov}(\Gamma_i, \Gamma_j)| \\
&= 2 \sum_{d=1}^{T-1} \sum_{\substack{1 \leq i < j \leq T \\ j-i=d}} |\mathrm{Cov}(\Gamma_i, \Gamma_j)| \\
&\leq 2 \sum_{d=1}^{T-1} (T-d) \left( 16\gamma^{\max(0, d-k-1)}(k+1)^2 M^2 \right) \\
&= 32 \left[ \sum_{d=1}^{k+1} (T-d)(k+1)^2 M^2 + \sum_{d=k+2}^{T-1} (T-d)(k+1)^2 M^2 \gamma^{d-k-1} \right] \\
&\leq 32 \left[ T(k+1)^3 M^2 + T(k+1)^2 M^2 \frac{\gamma}{1-\gamma} \right].
\end{aligned}$$

Finally, we conclude the proof by noting that

$$\begin{aligned}
\mathrm{Var}(\hat{\tau}_k) &= \frac{1}{T^2} \left[ \sum_{t=1}^{T} \mathrm{Var}(\Gamma_t) + 2 \sum_{1 \leq i < j \leq T} \mathrm{Cov}(\Gamma_i, \Gamma_j) \right] \\
&\leq \frac{1}{T} \left[ 4(k+1)^2 M^2 + 32(k+1)^3 M^2 + \frac{32\gamma(k+1)^2 M^2}{1-\gamma} \right] \\
&= O\left( \frac{(k+1)^3 M^2}{T} + \frac{\gamma(k+1)^2 M^2}{T(1-\gamma)} \right).
\end{aligned}$$

$\square$

### D.2. Proof of Theorem 6.3

*Lemma* D.1 (CLT for the TPG estimator). *Fix a finite truncation size $k \geq 0$. Under Assumptions 2.1 and 6.1, define*

$$\sigma_k^2 := \lim_{T \to \infty} \frac{1}{T} \mathrm{Var}\left( \sum_{u=1}^{T} \left( \frac{\mathbf{1}\{Z_u = 1\}}{1/2} - \frac{\mathbf{1}\{Z_u = 0\}}{1/2} \right) \sum_{t=u}^{\min(u+k, T)} Y_t \right),$$

*then, as $T \to \infty$, $\sqrt{T}\left( \hat{\tau}_k - \mathbb{E}[\hat{\tau}_k] \right) \Rightarrow \mathcal{N}(0, \sigma_k^2)$.*

*Lemma* D.2 (Consistency of the ideal HAC estimator). *Under Assumptions 2.1 and 6.1, define*

$$V_t := B_t - \mathbb{E}[B_t], \qquad \tilde{\Gamma}_\ell := \frac{1}{T}\sum_{t=1}^{T-\ell} V_t V_{t+\ell}, \qquad w_\ell^{\mathrm{NW}} := 1 - \frac{\ell}{L_T + 1},$$

*where $L_T \to \infty$ with $L_T = O(T^{1/3})$. Construct the ideal HAC estimator as*

$$\tilde{\Omega}_T := \tilde{\Gamma}_0 + 2\sum_{\ell=1}^{L_T} w_\ell^{\mathrm{NW}} \tilde{\Gamma}_\ell.$$

*Then, $\tilde{\Omega}_T$ is a consistent estimator of the asymptotic variance $\sigma_k^2$ in Lemma D.1, i.e., $\tilde{\Omega}_T \xrightarrow{p} \sigma_k^2$.*

*Lemma* D.3 (Consistency of the nonstationary HAC estimator under Cesàro-$L^2$ mean stability). *Under Assumption 2.1, 6.1, and 6.2, let $\bar{B} := T^{-1}\sum_{t=1}^{T} B_t$,*

$$\hat{V}_t := B_t - \bar{B}, \qquad \hat{\Gamma}_\ell := \frac{1}{T}\sum_{t=1}^{T-\ell} \hat{V}_t \hat{V}_{t+\ell}, \qquad w_\ell^{\mathrm{NW}} := 1 - \frac{\ell}{L_T + 1},$$

*where $L_T \to \infty$ with $L_T = O(T^{1/3})$. Construct the nonstationary HAC estimator as*

$$\hat{\Omega}_T := \hat{\Gamma}_0 + 2\sum_{\ell=1}^{L_T} w_\ell^{\mathrm{NW}} \hat{\Gamma}_\ell.$$

*Then, as $T \to \infty$, $\hat{\Omega}_T - \tilde{\Omega}_T = o_p(1)$.*

*Proof of Theorem 6.3.* The theorem follows from Lemma D.1, Lemma D.2, and Lemma D.3. □

# E. Proof of Propositions

### E.1. Proof of Proposition 4.1

*Proof of Proposition 4.1.* The value function $J(\theta)$ can be written as

$$J(\theta) = \frac{1}{T}\sum_{t=1}^{T} \mathbb{E}_{\mathcal{L}_\theta^t}[Y_t] = \frac{1}{T}\sum_{t=1}^{T} \rho^\top P_\theta^{1\to t} r_\theta.$$

Then, differentiating $J(\theta)$ yields

$$\frac{d}{d\theta}J(\theta) = \frac{1}{T}\sum_{t=1}^{T} \frac{d}{d\theta}\rho^\top P_\theta^{1\to t} r_\theta$$

$$= \frac{1}{T}\sum_{t=1}^{T} \rho^\top \left[\sum_{u=1}^{t-1} P_\theta^{1\to u}\Delta^u P_\theta^{(u+1)\to t}\right] r_\theta + \frac{1}{T}\sum_{t=1}^{T} \rho^\top P_\theta^{1\to t}(r_1 - r_0)$$

$$= \frac{1}{T}\sum_{t=1}^{T} \left[\rho^\top P_\theta^{1\to t}(r_1 - r_0) + \sum_{u=1}^{t-1} \rho^\top P_\theta^{1\to u}\Delta^u P_\theta^{(u+1)\to t} r_\theta\right].$$

Evaluate at $\theta = 1/2$ (more generally, for any $\theta \in (0,1)$), we have

$$\nabla J(1/2) = \frac{d}{d\theta}J(\theta)\bigg|_{\theta=1/2}$$

$$= \frac{1}{T}\sum_{t=1}^{T} \left(\sum_{u=1}^{t-1} \rho^\top P_{1/2}^{1\to u}\Delta^u P_{1/2}^{(u+1)\to t} r_{1/2} + \rho^\top P_{1/2}^{1\to t}(r_1 - r_0)\right).$$

Observe that each term $\rho^\top P_{1/2}^{1\to u} \Delta^u P_{1/2}^{(u+1)\to t} r_{1/2}$ corresponds to the difference in the conditional expected outcome at time $t$ between taking action $Z_u = 1$ versus $Z_u = 0$ at time $u$, under policy $\pi_{1/2}$ and initial state distribution $X_1 \sim \rho$. Thus, we can express the gradient as

$$\nabla J(1/2) = \frac{1}{T} \sum_{t=1}^{T} \sum_{u=1}^{t} \left( \mathbb{E}_{\mathcal{L}_{1/2}^t}[Y_t \mid Z_u = 1] - \mathbb{E}_{\mathcal{L}_{1/2}^t}[Y_t \mid Z_u = 0] \right)$$

$$= \frac{1}{T} \sum_{u=1}^{T} \sum_{t=u}^{T} \left( \mathbb{E}_{\mathcal{L}_{1/2}^t}[Y_t \mid Z_u = 1] - \mathbb{E}_{\mathcal{L}_{1/2}^t}[Y_t \mid Z_u = 0] \right)$$

$$= \frac{1}{T} \sum_{u=1}^{T} \left( Q_{1/2}^u(1) - Q_{1/2}^u(0) \right)$$

$$= \frac{1}{T} \sum_{t=1}^{T} \left( Q_{1/2}^t(1) - Q_{1/2}^t(0) \right),$$

which completes the proof. $\qquad\square$

### E.2. Proof of Proposition 4.2

*Proof of Proposition 4.2.* We first show that the untruncated PG estimator $\hat\tau_T$ is an unbiased estimator of the policy gradient $\nabla J(1/2)$. Recall that

$$\hat\tau_T = \frac{1}{T} \sum_{u=1}^{T} \left( \frac{\mathbf{1}\{Z_u = 1\}}{1/2} - \frac{\mathbf{1}\{Z_u = 0\}}{1/2} \right) \sum_{t=u}^{T} Y_t.$$

Under the experimental policy $\pi_{1/2}$ and initial state distribution $X_1 \sim \rho$, taking expectation gives

$$\mathbb{E}[\hat\tau_T] = \frac{1}{T} \sum_{u=1}^{T} \mathbb{E}\left[ \frac{\mathbf{1}\{Z_u = 1\}}{1/2} \sum_{t=u}^{T} Y_t - \frac{\mathbf{1}\{Z_u = 0\}}{1/2} \sum_{t=u}^{T} Y_t \right]$$

$$= \frac{1}{T} \sum_{u=1}^{T} \left( Q_{1/2}^u(1) - Q_{1/2}^u(0) \right) = \nabla J(1/2),$$

where the final equality follows from Proposition 4.1. Hence, $\hat\tau_T$ is an unbiased estimator of $\nabla J(1/2) = J'(1/2)$.

We then prove the bias bound. Recall that

$$J(\theta) := \frac{1}{T} \sum_{t=1}^{T} \rho^\top P_\theta^{1\to t} r_\theta, \qquad \tau = J(1) - J(0).$$

Since $\mathbb{E}[\hat\tau_T] = J'(1/2)$,

$$\mathbb{E}[\hat\tau_T] - \tau = J'(1/2) - \left( J(1) - J(0) \right).$$

Using the integral form of the Taylor remainder around $\theta = 1/2$,

$$\mathbb{E}[\hat\tau_T] - \tau = \int_0^{1/2} \theta J''(\theta)\, d\theta - \int_{1/2}^{1} (1-\theta) J''(\theta)\, d\theta. \qquad (26)$$

It remains to bound this centered remainder. Let

$$d_r := r_1 - r_0, \qquad \Delta^u := P_1^u - P_0^u, \qquad \bar\theta := 1/2.$$

By differentiating $J(\theta)$ twice, we may write

$$J''(\theta) = 2D(\theta) + A(\theta) + B(\theta), \qquad (27)$$

where

$$D(\theta) := \frac{1}{T} \sum_{t=1}^{T} \sum_{u=1}^{t-1} \rho^{\top} P_{\theta}^{1\to u} \Delta^{u} P_{\theta}^{(u+1)\to t} d_r, \tag{28}$$

$$A(\theta) := \frac{1}{T} \sum_{t=1}^{T} \sum_{u=1}^{t-1} \sum_{i=1}^{u-1} \rho^{\top} P_{\theta}^{1\to i} \Delta^{i} P_{\theta}^{(i+1)\to u} \Delta^{u} P_{\theta}^{(u+1)\to t} r_{\theta}, \tag{29}$$

$$B(\theta) := \frac{1}{T} \sum_{t=1}^{T} \sum_{u=1}^{t-1} \sum_{j=u+1}^{t-1} \rho^{\top} P_{\theta}^{1\to u} \Delta^{u} P_{\theta}^{(u+1)\to j} \Delta^{j} P_{\theta}^{(j+1)\to t} r_{\theta}. \tag{30}$$

Define the first-order term at the centered design by

$$\bar{D} := D(\bar{\theta}) = D(1/2).$$

Then

$$J''(\theta) = 2\bar{D} + R(\theta), \qquad R(\theta) := 2\big(D(\theta) - \bar{D}\big) + A(\theta) + B(\theta). \tag{31}$$

We now show that, without any mixing assumption,

$$\sup_{\theta \in [0,1]} |R(\theta)| = O\big(T^2 \delta^2 M\big).$$

We only use the non-expansiveness of Markov kernels in total variation:

$$\|\mu P_{\theta}^{a\to b}\|_{\mathrm{TV}} \leq \|\mu\|_{\mathrm{TV}},$$

together with

$$\|\mu \Delta^{u}\|_{\mathrm{TV}} \leq \delta \|\mu\|_{\mathrm{TV}}, \qquad \|r_{\theta}\|_{\infty} \leq M, \qquad \|d_r\|_{\infty} \leq 2M.$$

First, every term in $A(\theta)$ contains two $\Delta$ insertions. Therefore, uniformly over $\theta \in [0,1]$,

$$|A(\theta)| \leq \frac{\delta^2 M}{T} \sum_{t=1}^{T} \sum_{u=1}^{t-1} \sum_{i=1}^{u-1} 1 = O\big(T^2 \delta^2 M\big). \tag{32}$$

Similarly,

$$|B(\theta)| \leq \frac{\delta^2 M}{T} \sum_{t=1}^{T} \sum_{u=1}^{t-1} \sum_{j=u+1}^{t-1} 1 = O\big(T^2 \delta^2 M\big). \tag{33}$$

It remains to bound $D(\theta) - \bar{D}$. By adding and subtracting $P_{\bar{\theta}}^{1\to u} \Delta^{u} P_{\theta}^{(u+1)\to t}$,

$$D(\theta) - \bar{D} = \frac{1}{T} \sum_{t=1}^{T} \sum_{u=1}^{t-1} \rho^{\top} \left( P_{\theta}^{1\to u} - P_{\bar{\theta}}^{1\to u} \right) \Delta^{u} P_{\theta}^{(u+1)\to t} d_r$$

$$+ \frac{1}{T} \sum_{t=1}^{T} \sum_{u=1}^{t-1} \rho^{\top} P_{\bar{\theta}}^{1\to u} \Delta^{u} \left( P_{\theta}^{(u+1)\to t} - P_{\bar{\theta}}^{(u+1)\to t} \right) d_r. \tag{34}$$

For the first difference, the product telescoping identity gives

$$P_{\theta}^{1\to u} - P_{\bar{\theta}}^{1\to u} = \big(\theta - \bar{\theta}\big) \sum_{i=1}^{u-1} P_{\theta}^{1\to i} \Delta^{i} P_{\bar{\theta}}^{(i+1)\to u}.$$

For the second difference,

$$P_{\theta}^{(u+1)\to t} - P_{\bar{\theta}}^{(u+1)\to t} = \big(\theta - \bar{\theta}\big) \sum_{j=u+1}^{t-1} P_{\theta}^{(u+1)\to j} \Delta^{j} P_{\bar{\theta}}^{(j+1)\to t}.$$

Since $|\theta - \bar{\theta}| \le 1/2$, each term in (34) contains two $\Delta$ insertions. Hence, uniformly over $\theta \in [0,1]$,

$$\left| D(\theta) - \bar{D} \right| \le \frac{C\delta^2 M}{T} \sum_{t=1}^{T} \sum_{u=1}^{t-1} \left\{ \sum_{i=1}^{u-1} 1 + \sum_{j=u+1}^{t-1} 1 \right\}$$
$$= O\left( T^2 \delta^2 M \right), \tag{35}$$

for a universal constant $C > 0$.

Combining (32), (33), and (35) gives

$$\sup_{\theta \in [0,1]} |R(\theta)| = O\left( T^2 \delta^2 M \right).$$

Substituting the decomposition (31) into (26), we obtain

$$\mathbb{E}[\hat{\tau}_T] - \tau = 2\bar{D} \left( \int_0^{1/2} \theta \, d\theta - \int_{1/2}^{1} (1-\theta) \, d\theta \right) + \int_0^{1/2} \theta R(\theta) \, d\theta - \int_{1/2}^{1} (1-\theta) R(\theta) \, d\theta.$$

The first term is exactly zero because

$$\int_0^{1/2} \theta \, d\theta = \int_{1/2}^{1} (1-\theta) \, d\theta = \frac{1}{8}.$$

Therefore,

$$|\mathbb{E}[\hat{\tau}_T] - \tau| \le \left( \int_0^{1/2} \theta \, d\theta + \int_{1/2}^{1} (1-\theta) \, d\theta \right) \sup_{\theta \in [0,1]} |R(\theta)|$$
$$= O\left( T^2 \delta^2 M \right).$$

This proves the bias bound. Since the argument used only the identity $\mathbb{E}[\hat{\tau}] = J'(1/2)$, the same bound applies to any unbiased estimator of $\nabla J(1/2)$.

We now bound the variance of $\hat{\tau}_T$. By definition,

$$\hat{\tau}_T = \frac{1}{T} \sum_{u=1}^{T} \left( \frac{\mathbf{1}\{Z_u = 1\}}{1/2} - \frac{\mathbf{1}\{Z_u = 0\}}{1/2} \right) \sum_{t=u}^{T} Y_t.$$

Since

$$\left| \frac{\mathbf{1}\{Z_u = 1\}}{1/2} - \frac{\mathbf{1}\{Z_u = 0\}}{1/2} \right| \le 2, \qquad |Y_t| \le M,$$

we have

$$|\hat{\tau}_T| \le \frac{1}{T} \sum_{u=1}^{T} 2 \sum_{t=u}^{T} |Y_t|$$
$$\le \frac{2M}{T} \sum_{u=1}^{T} (T - u + 1) = M(T + 1).$$

Thus,

$$\mathrm{Var}(\hat{\tau}_T) \le \mathbb{E}\left[ \hat{\tau}_T^2 \right] \le M^2 (T + 1)^2 = O\left( T^2 M^2 \right).$$

This completes the proof. $\square$

### E.3. Proof of Proposition 5.1

*Proof of Proposition 5.1.* The truncated policy value function $J_k(\theta)$ can be written as

$$J_k(\theta) = \frac{1}{T} \sum_{t=1}^{T} \mathbb{E}_{\mathcal{L}_\theta^{t,k}}[Y_t] = \frac{1}{T} \sum_{t=1}^{T} \rho^\top P_{1/2}^{1 \to \max(t-k,1)} P_\theta^{\max(t-k,1) \to t} r_\theta.$$

Then, differentiating $J_k(\theta)$ yields

$$\frac{d}{d\theta} J_k(\theta)$$

$$= \frac{1}{T} \sum_{t=1}^{T} \frac{d}{d\theta} \rho^\top P_{1/2}^{1 \to \max(t-k,1)} P_\theta^{\max(t-k,1) \to t} r_\theta$$

$$= \frac{1}{T} \sum_{t=1}^{T} \rho^\top P_{1/2}^{1 \to \max(t-k,1)} \left[ \sum_{u=\max(t-k,1)}^{t-1} P_\theta^{\max(t-k,1) \to u} \Delta^u P_\theta^{(u+1) \to t} r_\theta \right.$$

$$\left. + P_\theta^{\max(t-k,1) \to t} (r_1 - r_0) \right].$$

Evaluate at $\theta = 1/2$, one can observe

$$\nabla J_k(1/2) = \left. \frac{d}{d\theta} J_k(\theta) \right|_{\theta=\frac{1}{2}}$$

$$= \frac{1}{T} \sum_{t=1}^{T} \left[ \rho^\top P_{1/2}^{1 \to t} (r_1 - r_0) + \sum_{u=\max(t-k,1)}^{t-1} \rho^\top P_{1/2}^{1 \to u} \Delta^u P_{1/2}^{(u+1) \to t} r_{1/2} \right]$$

$$= \frac{1}{T} \sum_{t=1}^{T} \sum_{u=\max(t-k,1)}^{t} \left( \mathbb{E}_{\mathcal{L}_{1/2}^t}[Y_t \mid Z_u = 1] - \mathbb{E}_{\mathcal{L}_{1/2}^t}[Y_t \mid Z_u = 0] \right)$$

$$= \frac{1}{T} \sum_{u=1}^{T} \sum_{t=u}^{\min(u+k,T)} \left( \mathbb{E}_{\mathcal{L}_{1/2}^t}[Y_t \mid Z_u = 1] - \mathbb{E}_{\mathcal{L}_{1/2}^t}[Y_t \mid Z_u = 0] \right)$$

$$= \frac{1}{T} \sum_{t=1}^{T} \left( Q_{1/2}^{t,k}(1) - Q_{1/2}^{t,k}(0) \right).$$

$\square$

## F. Proof of Lemmas

### F.1. Proof of Lemma D.1

*Definition* F.1 (Dobrushin, ergodic, and minimal ergodic coefficients). Let $K$ be a Markov transition kernel on a measurable state space $(\mathcal{X}, \mathcal{B})$. The *Dobrushin contraction coefficient* of $K$ is

$$\tilde{\gamma}(K) := \sup_{x_1, x_2 \in \mathcal{X}} \sup_{B \in \mathcal{B}} \left| K(x_1, B) - K(x_2, B) \right|,$$

and the associated *ergodic coefficient* is $\tilde{\alpha}(K) := 1 - \tilde{\gamma}(K)$. For each $n \geq 1$, let $\{X_{n,i} : 1 \leq i \leq N_n\}$ be a finite temporally nonhomogeneous Markov chain with one-step kernels $K_{i,i+1}^{(n)}(x, \cdot)$ satisfying

$$\mathbb{P}(X_{n,i+1} \in \cdot \mid X_{n,1}, \ldots, X_{n,i}) = K_{i,i+1}^{(n)}(X_{n,i}, \cdot).$$

The *minimal ergodic coefficient* of the $n$-th row is then

$$\tilde{\alpha}_n \;:=\; \min_{1 \le i < N_n} \tilde{\alpha}\left(K_{i,i+1}^{(n)}\right) \;=\; 1 - \max_{1 \le i < N_n} \tilde{\gamma}\left(K_{i,i+1}^{(n)}\right).$$

*Lemma* F.2 (CLT for temporally nonhomogeneous Markov chains in (Arlotto & Steele, 2016)). *Fix $m \ge 0$. For each $n \ge 1$, let $\{X_{n,i}\}_{i=1}^{n+m}$ be as in Definition F.1, and let $\{f_{n,i}\}_{i=1}^n$ be bounded measurable functions (i.e., $f_{n,i} : \mathcal{X}^{1+m} \to \mathbb{R}$) with*

$$S_n \;:=\; \sum_{i=1}^{n} f_{n,i}\left(X_{n,i}, \ldots, X_{n,i+m}\right), \qquad \max_{1 \le i \le n} \|f_{n,i}\|_\infty \;\le\; C_n,$$

*where $C_n$ may depend on $n$. Suppose the minimal ergodic coefficients satisfy $\tilde{\alpha}_n > 0$ and*

$$C_n^2 \, \tilde{\alpha}_n^{-2} \;=\; o\left(\mathrm{Var}(S_n)\right), \quad \text{as } n \to \infty.$$

*Then, we have*

$$\frac{S_n - \mathbb{E}[S_n]}{\sqrt{\mathrm{Var}(S_n)}} \;\Rightarrow\; \mathcal{N}(0,1), \quad \text{as } n \to \infty.$$

*Proof of Lemma D.1.* The proof follows by recasting our nonstationary MDP under the experimenting policy $\pi_{1/2}$ as a temporally nonhomogeneous Markov chain and verifying the conditions of Lemma F.2.

Let

$$G(x, z, u) := F_{R(\cdot|x,z)}^{-1}(u),$$

where $F_{R(\cdot|x,z)}^{-1}$ denotes a generalized inverse CDF. Using independent $U_t \sim \mathrm{Unif}[0,1]$, the reward can be represented as

$$Y_t = G(X_t, Z_t, U_t).$$

For each horizon $T$, define the shifted augmented row

$$\widetilde{X}_{T,t} := (X_t, Z_{t-1}, U_{t-1}), \qquad 1 \le t \le T+1,$$

where $Z_0$ and $U_0$ are arbitrary dummy variables independent of everything else. For $t > T+1$, extend the row to length $T + k + 1$ using any fixed transition kernel that is independent of the current state; the functions below do not use these additional variables.

For $1 \le t \le T$, the one-step transition kernel of $\widetilde{X}_{T,t}$ is, for $w = (x, z_-, u_-) \in \mathcal{X} \times \{0,1\} \times [0,1]$ and measurable $A \subseteq \mathcal{X} \times \{0,1\} \times [0,1]$,

$$\widetilde{K}_t(w, A) := \frac{1}{2} \sum_{z=0}^{1} \int_0^1 \sum_{y \in \mathcal{X}} \mathbf{1}\{(y, z, u) \in A\} P_z^t(x, y) \, du.$$

This shifted augmentation is Markov because, conditional on $X_t$, the fresh action $Z_t$ and reward noise $U_t$ are drawn independently, and then $X_{t+1}$ is generated from $P_{Z_t}^t(X_t, \cdot)$.

We now verify the Dobrushin coefficient of $\widetilde{K}_t$. Fix $w = (x, z_-, u_-)$ and $w' = (x', z'_-, u'_-)$, and for each $z, u$ define the section

$$A_{z,u} := \{y \in \mathcal{X} : (y, z, u) \in A\}.$$

Then, for $1 \le t \le T$,

$$\left|\widetilde{K}_t(w, A) - \widetilde{K}_t(w', A)\right| \le \frac{1}{2} \sum_{z=0}^{1} \int_0^1 \left|P_z^t(x, A_{z,u}) - P_z^t(x', A_{z,u})\right| du$$

$$\le \frac{1}{2} \sum_{z=0}^{1} \gamma \;=\; \gamma,$$

where the second inequality follows from Assumption 2.1 applied to point masses. For the artificially extended kernels after time $T$, the Dobrushin coefficient is zero because those kernels are independent of the current state. Hence the minimal ergodic coefficient of the row satisfies

$$\tilde{\alpha}_T \geq 1 - \gamma > 0.$$

We next define the local functions. For local arguments $w_\ell = (x_\ell, a_\ell, u_\ell) \in \mathcal{X} \times \{0, 1\} \times [0, 1], \ell = 0, \ldots, k + 1$, define

$$f_{T,i}(w_0, \ldots, w_{k+1}) := 2(2a_1 - 1) \sum_{j=0}^{k} \mathbf{1}\{i + j \leq T\} G(x_j, a_{j+1}, u_{j+1}).$$

Since $w_\ell$ corresponds to $\widetilde{X}_{T,i+\ell}$, this gives

$$f_{T,i}\left(\widetilde{X}_{T,i}, \ldots, \widetilde{X}_{T,i+k+1}\right) = 2(2Z_i - 1) \sum_{j=0}^{k} \mathbf{1}\{i + j \leq T\} Y_{i+j}.$$

Therefore, with

$$S_T := \sum_{i=1}^{T} f_{T,i}\left(\widetilde{X}_{T,i}, \ldots, \widetilde{X}_{T,i+k+1}\right),$$

we have $S_T = T\hat{\tau}_k$.

We now verify the remaining conditions of Lemma F.2, applying it with $n = T$ and $m = k + 1$. Since $|Y_t| \leq M$ almost surely,

$$\max_{1 \leq i \leq T} \|f_{T,i}\|_\infty \leq 2(k + 1)M.$$

Moreover, by the assumed existence of the asymptotic variance,

$$\frac{1}{T} \operatorname{Var}(S_T) \to \sigma_k^2 > 0.$$

Then $\operatorname{Var}(S_T) \sim T\sigma_k^2 \to \infty$, and

$$C_T^2 \tilde{\alpha}_T^{-2} \leq 4(k + 1)^2 M^2 (1 - \gamma)^{-2} = o(\operatorname{Var}(S_T)).$$

Lemma F.2 therefore yields

$$\frac{S_T - \mathbb{E}[S_T]}{\sqrt{\operatorname{Var}(S_T)}} \Rightarrow \mathcal{N}(0, 1).$$

Since $\operatorname{Var}(S_T)/T \to \sigma_k^2$, Slutsky's theorem gives

$$\sqrt{T}\left(\hat{\tau}_k - \mathbb{E}[\hat{\tau}_k]\right) = \frac{S_T - \mathbb{E}[S_T]}{\sqrt{\operatorname{Var}(S_T)}} \sqrt{\frac{\operatorname{Var}(S_T)}{T}} \Rightarrow \mathcal{N}(0, \sigma_k^2).$$

$\square$

## F.2. Proof of Lemma D.2

*Proof of Lemma D.2.* After reindexing, we can write

$$\hat{\tau}_k = \frac{1}{T} \sum_{u=1}^{T} w_u^{\text{IPW}} \sum_{t=u}^{\min(u+k, T)} Y_t = \frac{1}{T} \sum_{t=1}^{T} \sum_{i=\max\{1, t-k\}}^{t} w_i^{\text{IPW}} Y_t = \frac{1}{T} \sum_{t=1}^{T} B_t.$$

Set the target variance as

$$\Omega := \lim_{T \to \infty} \frac{1}{T} \sum_{i=1}^{T} \sum_{j=1}^{T} \mathbb{E}[V_i V_j].$$

Then

$$\Omega \;=\; \lim_{T\to\infty}\frac{1}{T}\mathbb{E}\Big[\Big(\textstyle\sum_{t=1}^{T}V_t\Big)\Big(\textstyle\sum_{s=1}^{T}V_s\Big)\Big] \;=\; \lim_{T\to\infty}\frac{1}{T}\,\mathrm{Var}\Big(\sum_{t=1}^{T}B_t\Big),$$

which is the *asymptotic variance* in the CLT of Lemma D.1, i.e., $\Omega = \sigma_k^2$.

For lags $\ell \geq 0$, form the *ideal* autocovariances

$$\tilde{\Gamma}_\ell \;:=\; \frac{1}{T}\sum_{t=1}^{T-\ell} V_t\, V_{t+\ell} \quad (\text{Define } \tilde{\Gamma}_{-\ell} := \tilde{\Gamma}_\ell).$$

With a bandwidth $L_T \to \infty$ and $L_T = O(T^{1/3})$ and Bartlett weights $w_\ell^{\mathrm{NW}} = 1 - \ell/(L_T + 1)$, define the ideal HAC estimate of the *asymptotic variance* of $\sqrt{T}(\hat{\tau}_k - \mathbb{E}[\hat{\tau}_k])$:

$$\tilde{\Omega}_T \;:=\; \sum_{j=-T}^{T} k\,(j/S_T)\,\tilde{\Gamma}_j \;=\; \tilde{\Gamma}_0 \;+\; 2\sum_{\ell=1}^{L_T} w_\ell^{\mathrm{NW}}\,\tilde{\Gamma}_\ell,$$

where $k(\cdot)$ is specified as the Bartlett kernel, i.e., $k(x) = (1 - |x|)\mathbf{1}\{|x| \leq 1\}$, and $S_T := L_T + 1$ is the bandwidth parameter. We verify the following conditions to apply Theorem 1 in Hansen (1992).

(K) For all $x \in \mathbb{R}, |k(x)| \leqslant 1$, $k(x) = k(-x)$; $k(0) = 1$; $k(x)$ is continuous at zero; and for almost all $x \in \mathbb{R}$, $\int_{\mathbb{R}} |k(x)|dx < \infty$.

(S) $S_T \to \infty$ and for some $q \in (1/2, \infty)$, $S_T^{1+2q}/T = O(1)$.

(V1) For some $r \in (2, 4]$ such that $r > 2 + 1/q$, and some $p > r$, $12\sum_{h=1}^{\infty} \alpha(h)^{2(1/r - 1/p)} = A < \infty$ and $\sup_{t\geqslant 1}\|V_t\|_p = C < \infty$, where $\alpha(\cdot)$ is the strong mixing coefficients for $\{V_t\}_{t=1}^{T}$.

Note that condition (K) is automatically satisfied under the Bartlett kernel, while condition (S) holds when taking $q = 1$ and choosing the bandwidth parameter $L_T = O(T^{1/3})$. We now proceed to verify (V1) in the following.

For any $p \geq 1$, we have $\sup_{t\geq 1}\|V_t\|_p \leq 4(k+1)M < \infty$. Choose any $r \in (2, 4]$ with $r > 2 + 1/q$ (e.g., for Bartlett $q = 1$, take $r = 4$), and then take any $p > r$. Let $\beta := 2(1/r - 1/p) > 0$.

Recall that $B_t = \sum_{i=\max\{1,\,t-k\}}^{t} w_i^{\mathrm{IPW}} Y_t$. Lemma A.2 establishes that the process $\{(X_t, Z_t, Y_t)\}_{t=1}^{T}$ is strong mixing with coefficient $\alpha(h) = \gamma^{h-1}$. By a similar argument as in Corollary A.3, the sequence $\{B_t\}_{t=1}^{T}$ is also strong mixing, with coefficient $\alpha(h) \leq \gamma^{\max\{0,\,h-k-1\}}$. It follows that

$$\sum_{h=1}^{\infty} \alpha(h)^\beta = \sum_{h=1}^{k+1} \alpha(h)^\beta + \sum_{h=k+2}^{\infty} \alpha(h)^\beta \leq (k+1)\cdot 1 + \sum_{h=k+2}^{\infty} \left(\gamma^{h-k-1}\right)^\beta.$$

The tail is geometric with ratio $\gamma^\beta \in (0, 1)$, so

$$\sum_{h=k+2}^{\infty} \left(\gamma^{h-k-1}\right)^\beta = \frac{\gamma^\beta}{1 - \gamma^\beta} < \infty$$

Therefore, for any finite $k < \infty$,

$$12\sum_{h=1}^{\infty} \alpha(h)^{2(1/r-1/p)} \leq 12\left[(k+1) + \frac{\gamma^\beta}{1-\gamma^\beta}\right] < \infty,$$

which verifies (V1).

Together with the CLT established in Lemma D.1, Theorem 1 of Hansen (1992) implies that the ideal HAC estimator consistently estimates the asymptotic variance, i.e., $\tilde{\Omega}_T \overset{p}{\to} \sigma_k^2$. $\qquad\square$

## F.3. Proof of Lemma D.3

*Proof of Lemma D.3.* Recall that $\bar{B} := T^{-1} \sum_{t=1}^{T} B_t$, $\mu_t := \mathbb{E}[B_t]$, and $\bar{\mu}_T := T^{-1} \sum_{t=1}^{T} \mu_t$. Write

$$\bar{V}_T := \frac{1}{T} \sum_{t=1}^{T} V_t = \bar{B} - \bar{\mu}_T, \qquad \delta_t := \bar{\mu}_T - \mu_t.$$

Then

$$\hat{V}_t = B_t - \bar{B} = (B_t - \mu_t) - (\bar{B} - \bar{\mu}_T) - (\bar{\mu}_T - \mu_t) = V_t - \bar{V}_T - \delta_t.$$

Fix $\ell \geq 0$. Expanding $\hat{\Gamma}_\ell - \tilde{\Gamma}_\ell$ gives

$$\hat{\Gamma}_\ell - \tilde{\Gamma}_\ell = \frac{1}{T} \sum_{t=1}^{T-\ell} \left[ (V_t - \bar{V}_T - \delta_t)(V_{t+\ell} - \bar{V}_T - \delta_{t+\ell}) - V_t V_{t+\ell} \right]$$

$$= \underbrace{-\frac{\bar{V}_T}{T} \sum_{t=1}^{T-\ell} V_t - \frac{\bar{V}_T}{T} \sum_{t=1}^{T-\ell} V_{t+\ell} + \frac{T-\ell}{T} \bar{V}_T^2}_{=:A_\ell} \tag{36}$$

$$\underbrace{-\frac{1}{T} \sum_{t=1}^{T-\ell} \delta_t \, V_{t+\ell} - \frac{1}{T} \sum_{t=1}^{T-\ell} \delta_{t+\ell} \, V_t + \frac{\bar{V}_T}{T} \sum_{t=1}^{T-\ell} (\delta_t + \delta_{t+\ell})}_{=:B_\ell} \tag{37}$$

$$+ \underbrace{\frac{1}{T} \sum_{t=1}^{T-\ell} \delta_t \, \delta_{t+\ell}}_{=:C_\ell} . \tag{38}$$

We bound these uniformly over $0 \leq \ell \leq L_T$.

*(i) Bound for $A_\ell$.* Since $\sqrt{T} \, \bar{V}_T = O_p(1)$ by Lemma D.1 and $|V_t| \leq 4(k+1)M$ deterministically for any $0 \leq t \leq T$,

$$\frac{1}{T} \sum_{t=1}^{T-\ell} V_t = \bar{V}_T + O\left(\frac{\ell}{T}\right), \qquad \frac{1}{T} \sum_{t=1}^{T-\ell} V_{t+\ell} = \bar{V}_T + O\left(\frac{\ell}{T}\right),$$

hence $A_\ell = O_p\left(T^{-1}\right) + O_p\left(\frac{L_T}{T^{3/2}}\right) + O_p\left(\frac{L_T}{T^2}\right)$. Since $L_T = O(T^{1/3})$, $\sup_{\ell \leq L_T} |A_\ell| = O_p(T^{-1})$.

*(ii) Bound for $B_\ell$.* For the two mixed terms use Cauchy–Schwarz and $\sum_{t=1}^{T} V_t^2 \leq 16(k+1)^2 M^2 T$:

$$\left| \frac{1}{T} \sum_{t=1}^{T-\ell} \delta_t V_{t+\ell} \right| \leq \frac{1}{T} \left( \sum_{t=1}^{T} \delta_t^2 \right)^{1/2} \left( \sum_{t=1}^{T} V_t^2 \right)^{1/2} \leq 4(k+1)M \, \Delta_T,$$

and similarly for $\frac{1}{T} \sum \delta_{t+\ell} V_t$. For the part with $\bar{V}_T$,

$$\left| \frac{\bar{V}_T}{T} \sum_{t=1}^{T-\ell} (\delta_t + \delta_{t+\ell}) \right| \leq \frac{|\bar{V}_T|}{T} \left( \left| \sum_{t=1}^{T-\ell} \delta_t \right| + \left| \sum_{t=1}^{T-\ell} \delta_{t+\ell} \right| \right).$$

Because $\sum_{t=1}^{T} \delta_t = 0$ and $|\delta_t| \leq 4(k+1)M < \infty$, each partial sum is $O(L_T)$, so this term is $O_p\left(\frac{L_T}{T^{3/2}}\right)$. Therefore,

$$\sup_{\ell \leq L_T} |B_\ell| \leq 8(k+1)M \, \Delta_T + O_p\left(\frac{L_T}{T^{3/2}}\right) = O_p(\Delta_T) + O_p\left(T^{-7/6}\right).$$

*(iii) Bound for $C_\ell$.* Again by Cauchy–Schwarz and $\sum_{t=1}^{T-\ell} \delta_{t+\ell}^2 \leq \sum_{t=1}^{T} \delta_t^2$,

$$|C_\ell| \leq \frac{1}{T} \left( \sum_{t=1}^{T} \delta_t^2 \right)^{1/2} \left( \sum_{t=1}^{T} \delta_t^2 \right)^{1/2} = \Delta_T^2,$$

so $\sup_{\ell \le L_T} |C_\ell| = O_p(\Delta_T^2)$.

Combining (i)–(iii), uniformly for $0 \le \ell \le L_T$,

$$\hat{\Gamma}_\ell - \tilde{\Gamma}_\ell = O_p(T^{-1}) + O_p(\Delta_T) + O_p(\Delta_T^2).$$

Using $0 \le w_\ell^{\mathrm{NW}} \le 1$,

$$\hat{\Omega}_T - \tilde{\Omega}_T = (\hat{\Gamma}_0 - \tilde{\Gamma}_0) + 2\sum_{\ell=1}^{L_T} w_\ell^{\mathrm{NW}}(\hat{\Gamma}_\ell - \tilde{\Gamma}_\ell) = O_p\!\left(\frac{L_T}{T}\right) + O_p(L_T\Delta_T) + O_p(L_T\Delta_T^2).$$

Together with $L_T = O(T^{1/3})$ and $\Delta_T^2 = o(T^{-2/3})$ in Assumption 6.2, we have $L_T/T \to 0$, $L_T\Delta_T \to 0$, and $L_T\Delta_T^2 \to 0$. Hence $\hat{\Omega}_T - \tilde{\Omega}_T = o_p(1)$. $\qquad\square$

# G. Experimental Details

This section provides a detailed description of the simulation setup for Section 7; additional implementation details are available in the accompanying code. All experiments were run on a PC with a 12-core CPU and 18 GB of RAM.

## G.1. Two-State Nonstationary MDP

We simulate a two-state nonstationary MDP to evaluate the bias and variance of the TPG estimator under mixing rates $\gamma$ ranging in $[0, 1]$. All experiments are repeated over 1,000 independent trials with a horizon of $T = 5000$. For each trial, we simulate a treatment trajectory, a control trajectory, and an experimental trajectory. The ground truth $\tau$ is computed from the treatment and control trajectories, and the TPG estimator is evaluated on the experimental trajectory. Further details on the simulation are provided in Appendix G.1.

In Table 3, we observe that the DM estimator (i.e., $k = 0$) exhibits substantial bias, with a mean absolute error (MAE) approaching 50%. The untruncated PG estimator ($k = T$) exhibits such high bias that the sign is incorrect; and very high variance as well. In contrast, the TPG estimator substantially reduces the bias even with $k = 1$ while maintaining low variance. Figure 4 further illustrates the MAE across different truncation sizes and mixing rates; the optimal choice of $k$ can vary with $\gamma$, but remains small relative to $T = 5000$.

| $k$ | MAE (%) | STD |
|---|---|---|
| 0 (DM) | 50.400 | 0.126 |
| 1 | 30.05 | 0.243 |
| 3 | 24.97 | 0.465 |
| 5 | 29.12 | 0.684 |
| 10 | 47.20 | 1.210 |
| 50 | 206.09 | 5.328 |
| 100 | 402.77 | 10.470 |
| $T$ (PG) | 11332.40 | 300.064 |

*Table 3.* Evaluation of $\hat{\tau}_k$ across truncation sizes $k$, averaged over varying mixing rates $\gamma$. Reported metrics: MAE (mean absolute error, in %) and STD (standard deviation). Average true ATE: $\tau = 2.14$.

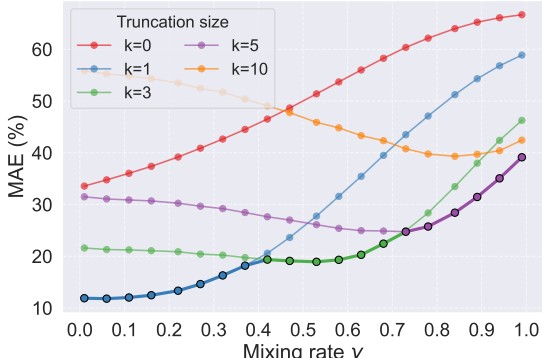

*Figure 4.* MAE of $\hat{\tau}_k$ with varying mixing rates $\gamma$. The optimal truncation size $k$ for each $\gamma$ is bolded.

In the experiment with a two-state non-stationary MDP, we consider a finite-horizon setting with state space $\mathcal{X} = \{0, 1\}$, action space $\mathcal{Z} = \{0, 1\}$, and horizon length $T = 5000$. The environment evolves according to time-varying transition kernels $P_z^t$ for $z \in \{0, 1\}$ and $t \in [T]$, which are constructed to model non-stationarity through a mean-reverting autoregressive process with additive i.i.d. Gaussian noise. Specifically, for each state $x \in \{0, 1\}$, the control transition kernel at time $t > 1$ is given by $P_0^t(x, \cdot) \propto \alpha P_0^{t-1}(x, \cdot) + (1 - \alpha)\mu_x + \varepsilon$, where $\alpha \in [0, 1]$ denotes the mean reversion rate, $\mu_x$ is both the long-run and the initial distribution of next-state transitions from state $x$, s.t. $\|\mu_0 - \mu_1\|_{\mathrm{TV}} = \gamma$, and $\varepsilon \sim \mathcal{N}(0, \sigma_\epsilon^2)$

denotes Gaussian noise. The treatment transition kernel is a shifted version of the control kernel, moving probability toward state 1, i.e., $P_1^t(x,1) \propto P_0^t(x,1) + \delta$ and $P_1^t(x,0) \propto P_0^t(x,0) - \delta$. Rewards are generated from a Normal distribution, i.e., $Y_t \sim \mathcal{N}(r(X_t, Z_t), \sigma_r^2)$.

In the simulation, we set the mean reversion rate to $\alpha = 0.5$, the standard deviations of the Gaussian noise and reward to $\sigma_\epsilon = \sigma_r = 0.1$ and the kernel deviation $\delta = 0.1$. The initial distribution $\mu_x$ is uniform randomly chosen to be either $[\frac{1}{2}(1+\gamma), \frac{1}{2}(1-\gamma)]$ or $[\frac{1}{2}(1-\gamma), \frac{1}{2}(1+\gamma)]$ (one for each $x \in \{0,1\}$), ensuring a total variation distance of $\gamma$ between $\mu_0$ and $\mu_1$. Mean rewards $r(x,z)$ are randomly generated for each state-action pair $(x,z)$. All experiments are repeated over 1,000 independent trials. Figure 5 further shows some examples under different kernel deviations $\delta$, signs-positive (panels a-h) and negative (panels i-p).

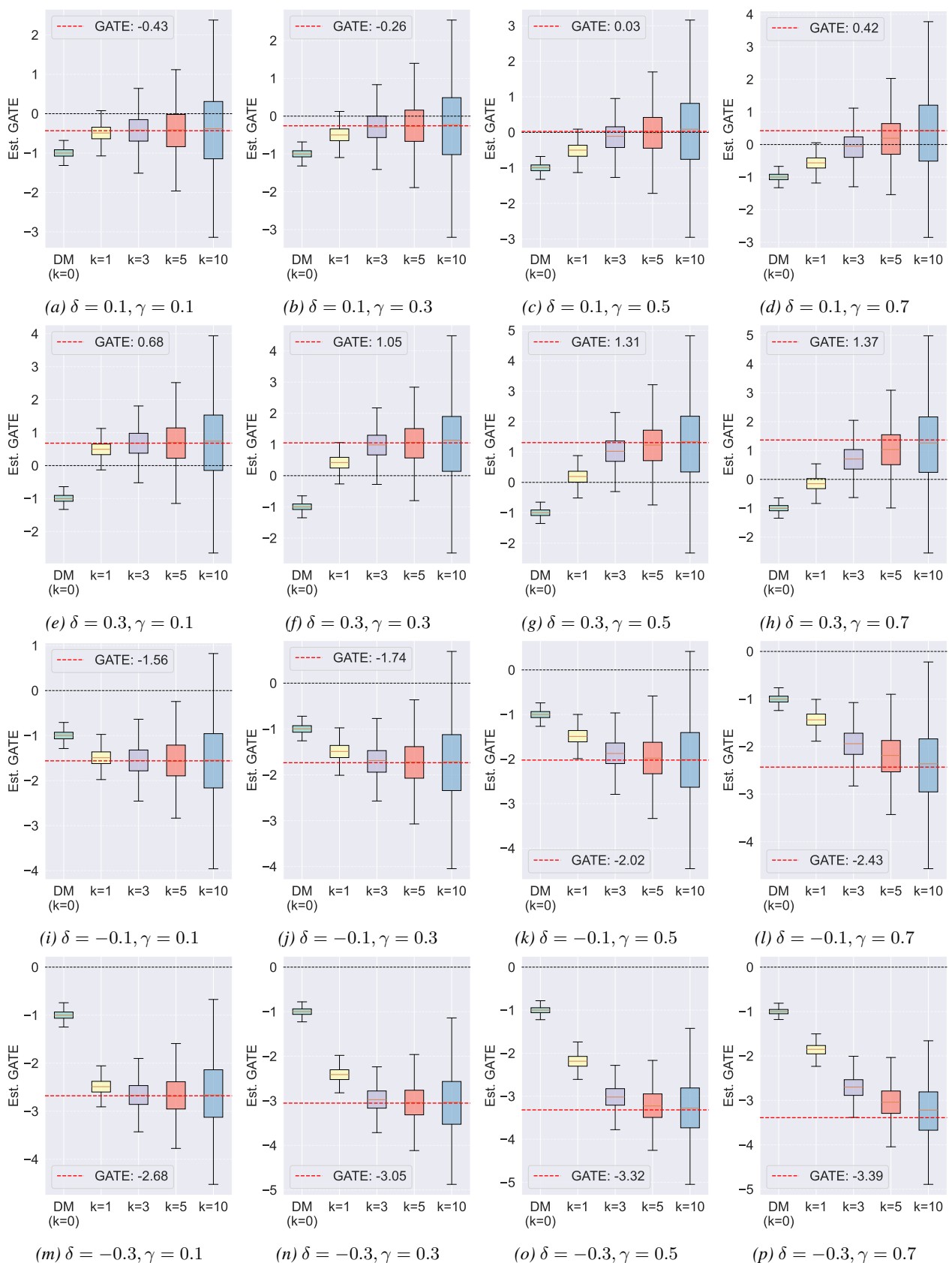

*Figure 5.* Evaluation of TPG estimators with various $\delta$ and $\gamma$ in a 2-state MDP.

### G.2. Nonstationary Queueing Simulator

In this section we detail our nonstationary queueing simulator, based on a nonstationary estimate of patient arrival rates to an emergency department (both based on day of week, and time of day), obtained using data from the SEEStat Home Hospital database (SEE-Center, 2013); see Figure 9, Section 5.2, and Appendix B.1 of (Li et al., 2023) for further details.

The patient arrival rates are estimated using data from the Home Hospital database on admissions to an Israeli hospital emergency department over 2004, for every half hour of each day, at the day-of-week level (Sunday-Saturday). Let $b_{d,t}$ denote this estimated arrival rate at time $t$, on a day of week $d$. We simulate a four-week experiment, similar to (Li et al., 2023). The arrival rate to the queueing system in week $w = 1, 2, 3, 4$, on day of week $d$, and time $t$, in state $k$, is specified as:

$$\lambda_{w,d,t,k} = \frac{8 - 4p}{1 + k/5} a_w b_{d,t},$$

where $a_1 = 0.9, a_2 = 1.0, a_3 = 1.1, a_4 = 1.2$. The service rate is $\mu = 20$ in all states. In treatment, $p = 1.75$; in control, $p = 0.25$.

To simulate the system and map it to the discrete-time setting in our paper, we *uniformize* time by considering one-minute time steps. This gives rise to $T = 40,320$ time steps per simulation. We run 500 simulations of each type of experiment (Bernoulli randomized experiment, and switchback experiment). In the switchback experiment, we set the switchback interval to one hour (i.e., 60 time steps), and switch between treatment and control with probability $1/2$ at each interval, following the Bernoulli switchback design of (Hu & Wager, 2022).

### G.3. NYC Ride-Sharing Simulator

We adopt a variant of the large-scale NYC ride-sharing simulator from (Peng et al., 2025) for the pricing experiment. The system observes a sequence of "eyeballs" (i.e., potential ride requests), where each rider specifies pickup and dropoff locations and is shown a posted price along with an estimated time of arrival (ETA). Based on this information, the rider decides whether to accept the trip. If accepted, the nearest available vehicle is dispatched and the system collects the trip fare as a reward; if rejected, no dispatch occurs and the reward is zero. The rider's acceptance behavior follows a simple logistic choice model:

$$\mathbb{P}(\text{accept}) = \sigma\left(w_{\text{price}} \cdot \text{price} + w_{\text{eta}} \cdot \text{ETA} + w_0\right),$$

where $\sigma(\cdot)$ is the sigmoid function. Specifically, we set $w_{\text{price}} = -0.3, w_{\text{eta}} = -0.005$, and $w_0 = 4$. The price is computed as price $= \alpha \cdot \tau(\text{pickup location}, \text{dropoff location})$, where $\tau$ denotes the travel time matrix derived from real Manhattan traffic data. The ETA is computed as the driver's earliest arrival time at the pickup location minus the request time. The pricing coefficient $\alpha$ is set to $0.01$ under the control policy and $0.02$ under the treatment policy. A similar modeling setup is described in Section 6.3 of (Peng et al., 2025). As shown in Figure 6, rider arrivals in the NYC ride-sharing data from September 2024 exhibit complex, non-stationary patterns (TLC, 2024), presenting a significant challenge for treatment effect estimation.

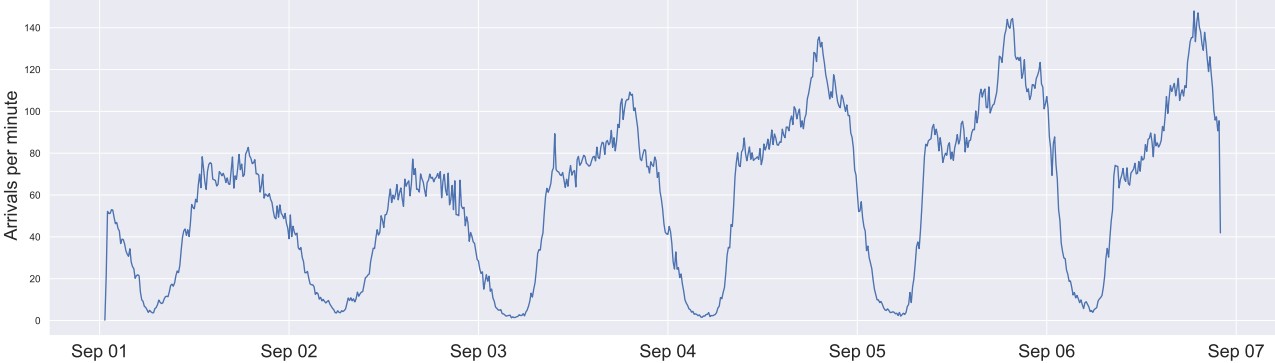

*Figure 6.* Nonstationary arrival rate (per minute) in the NYC ride-sharing dataset.

