# OpenReview forum: "Estimation of Treatment Effects Under Nonstationarity via the Truncated Policy Gradient Estimator"
_ICML.cc/2026/Conference — ICML 2026 regular_

### Official Review · Reviewer_KaTY · 2026-02-23

**Soundness:** 3
**Presentation:** 2
**Significance:** 2
**Originality:** 3
**Overall Recommendation:** 5
**Confidence:** 4

**Summary:**

This paper studies the problem of global average treatment effect in A/B tests with Markovian interference. This paper extends the scenario to a nonstationary environment (state transition kernel) and develops a state-agnostic estimator to construct a novel bias-variance tradeoff spectrum. Specifically, the authors substitute the direct rewards in the classical Difference-in-Means (DM) estimator with the sum of rewards in the next $k$ time steps and derive their Truncated Policy Gradient (TPG) estimator, where policy gradient corresponds to the first-order approximation to policy value. The authors then derive the order of bias and variance of the TPG estimator and introduce a hypothetical truncated policy to demonstrate its intuition with Taylor expansion. Next, the authors construct a CLT and a robust variance estimator. Finally, two experiments with realistic settings are conducted to demonstrate the advantage of the TPG estimator.

**Compliance With Llm Reviewing Policy:**

Affirmed.

**Final Justification:**

I finally raise my overall recommendation from 4 to 5. The paper is technically solid, well motivated, and studies a meaningful problem in treatment effect estimation under nonstationary Markovian interference. I find the main contribution valuable: the proposed TPG estimator is well supported by theory, and the empirical results suggest that it offers a useful and robust practical alternative in challenging settings.

My main earlier concerns were about the intuition for the estimator, the comparison with DM, and when TPG provides a concrete advantage. The rebuttal and follow-up reply addressed these concerns satisfactorily by clarifying the bias-variance tradeoff, giving more concrete regimes where truncation is beneficial, and providing a more principled interpretation of the estimator. These clarifications improved my confidence in both the method and its significance.

Overall, I view this as a solid and worthwhile contribution. While the presentation and intuition could still be improved in the final version, the core methodology and technical results are strong enough to merit acceptance.

**Key Questions For Authors:**

1. Can the authors further explain the intuition of the TPG estimator? I think Q-value in the Difference-in-Q estimator is a straightforward surrogate for direct reward in the DM estimator. However, it is hard for me to understand why the sum of rewards in the next $k$ steps is used to replace the direct reward in this paper, instead of a natural average or other aggregation method of future rewards (especially because the sum would influence the scale of the estimator, intuitively). Even though Section 5 interprets the TPG estimator as the first-order approximation of a hypothetical truncated policy, I think this hypothetical truncated policy itself is somewhat strange and does not provide an appropriate intuition for understanding.

2. Can the authors explain why the TPG estimator achieves effective policy evaluation through only a single sample path? i.e., as claimed in the main paper, ''*It can be computed from a single sample path, without requiring multiple i.i.d. trajectories*'', especially since multiple i.i.d. trajectories are generally needed for most other estimators. It seems a magic and not sufficiently revealed.

**Limitations:**

Yes.

**Strengths And Weaknesses:**

Strengths:
1. This paper is well-written with a clear structure.
2. The proposed method, Truncated Policy Gradient (TPG) estimator, is well-motivated and performs well in challenging nonstationary Markovian environments.
3. The theory of this paper is solid and presented clearly.

Weaknesses:
1. The theoretical comparison with the Difference-in-Means (DM) estimator and TPG estimator is missing, which makes the presentation in Section 3 a bit confusing. In the flow of Section 3, it is straightforward to take the TPG estimator as an extension of the DM estimator (just like in the Difference-in-Q (DQ) estimator, the direct reward is substituted with the estimated Q-value to tackle Markovian interference). However, the authors take TPG as an extension of the DM estimator, while no comparison (e.g., on the order of MSE) with the DM estimator is provided. Moreover, in Sections 4 and 5, the main baseline is policy gradient, which is a very weak one, and DM remains missing.

2. The empirical advantage of the TPG estimator is not significant. In the experiments, especially patient arrivals (the top one of Figure 2), with optimal tuning parameter (e.g., TPG(k=5) and BC(b=20)), I observe that the advantage of TPG estimator is not really significant. I think at least more analysis and explanation of the results are needed.

---

> ### Author Rebuttal · Authors · 2026-03-31
>
> **We thank the reviewer for their careful reading and constructive comments.**
>
> **Q1.** *The theoretical comparison with the Difference-in-Means (DM) estimator and TPG estimator is missing*.
>
> We note that DM is directly nested within the TPG framework: Eq. (2) reduces exactly to DM when $k=0$, and Figure 2 reports it explicitly. For this reason, Theorem 3.2 already gives the direct theoretical comparison the reviewer asks for. Setting $k=0$ yields the DM bias bound $O(\delta M/(1-\gamma))$ and variance $O(M^2/T + \gamma M^2/[T(1-\gamma)])$. For general k, the TPG bounds become $O(k^2 \delta^2 M + \gamma^k \delta M/(1-\gamma))$ for bias and $O((k+1)^3 M^2/T + \gamma (k+1)^2 M^2/[T(1-\gamma)])$ for variance. This makes the tradeoff explicit, i.e., increasing $k$ replaces the full omitted carryover by a geometrically decaying tail $\gamma^k/(1-\gamma)$, at the price of a larger truncation term and higher variance. We will explicitly emphasize this comparison in the camera ready version of the paper.
>
> **Q2.**  *The empirical advantage of the TPG estimator is not significant.*
>
> We agree that in Case Study I, a well-tuned BC baseline (e.g., $b=20$) can be competitive with TPG. Our claim is not that TPG uniformly dominates all carefully tuned switchback estimators. Rather, TPG is designed as a robust post-processing estimator that can be applied under a standard Bernoulli randomized experiment, whereas BC relies on a more restrictive switchback design together with burn-in tuning.
>
> The empirical advantage of TPG in Figure~2 lies primarily in its robustness. In particular, TPG consistently exhibits lower bias than DM across different interval lengths, while DM is highly sensitive to the choice of interval, and BC can exhibit overestimation when the burn-in parameter is misspecified. We will clarify this point and expand the discussion of these trade-offs in the camera ready version.
>
> **Q3.** *Can the authors further explain the intuition of the TPG estimator?*
>
> The key object captured by the TPG estimator is the \emph{downstream marginal effect} of a single treatment assignment. When action $Z_u$ affects the state dynamics, its impact propagates beyond the immediate reward $Y_u$ to future outcomes $Y_{u+1}, Y_{u+2}, \ldots$, through the underlying state evolving. Thus, focusing only on $Y_u$ (as in DM) misses this carryover effect.
>
> The quantity $\sum_{t=u}^{u+k} Y_t$ serves as a natural first-order surrogate for the total short-horizon impact of the same treatment assignment (i.e., the truncated Q-value). This choice is not arbitrary: Proposition 5.1 shows that $\mathbb{E}[\hat{\tau}_k]$ coincides with the gradient of the truncated policy value $J_k$ at $\theta = 1/2$. In other words, the estimator is calibrated to match a well-defined causal target, i.e., the marginal effect of perturbing the treatment probability on cumulative rewards over the next $k$ steps.
>
> Using an average (or other aggregation) within the window would simply rescale this quantity and, importantly, *would no longer align with the policy-gradient identity on the correct target scale*. The overall normalization is already handled by the outer $1/T$, so the sum form preserves the correct magnitude and interpretation of the effect.
>
> **Q4.** *Can the authors explain why the TPG estimator achieves effective policy evaluation through only a single sample path?*
>
> The ability to use a single trajectory is not ``magic,'' but rather follows from repeated randomized perturbations within the same path. At each time $u$, the Bernoulli randomization ensures that $Z_u$ is independent of the past. As a result, the weighted term
> $$
> \left(\frac{\mathbb{1}(Z_u=1)}{0.5} - \frac{\mathbb{1}(Z_u=0)}{0.5}\right) \sum_{t=u}^{u+k} Y_t
> $$
> has a conditional expectation equal to the difference between taking action $1$ versus $0$ at time $u$, followed by the experimental policy thereafter. Averaging over $u$ then recovers the truncated Q-value, i.e., the truncated policy gradient in expectation.
>
> Thus, statistical identification is achieved via time-indexed randomization within a single trajectory, rather than across independent trajectories. While multiple i.i.d. trajectories can help reduce variance (i.e., by taking the mean of the estimates), they are not required for identification.

---

> > ### Author Rebuttal · Reviewer_KaTY · 2026-04-01
> >
> > I would like to thank the authors for their detailed responses.
> >
> > While the clarification that DM is recovered as the special case $k=0$ is helpful, it does not fully address the core concern: the rebuttal establishes a bias–variance tradeoff but does not clearly characterize when TPG provides a concrete improvement over DM in a practically meaningful or verifiable regime. As such, the comparison remains largely qualitative.
> >
> > Regarding intuition, although the policy-gradient perspective is now clearer, it still appears **somewhat post hoc**: the truncated policy and $k$-step reward construction seem primarily introduced to align with a Taylor expansion and gradient identity, rather than arising naturally from the original estimand. This makes the method feel more like a carefully engineered approximation than a principled derivation.
> >
> > Overall, while I find the paper technically solid and the rebuttal helpful, I remain unconvinced that the proposed approach represents a sufficiently principled advance, and therefore I keep my original score.

---

> > > ### Author Response · Authors · 2026-04-03
> > >
> > > We thank the reviewer for the response.
> > >
> > > **Concrete Regimes**
> > >
> > > We now understand the reviewer’s question regarding the specific regimes in which truncation is beneficial. From the bias bounds for $k=0$ and $k\geq 1$, a concrete sufficient condition is
> > > $$(1-\gamma)k^2\delta + \gamma^k \leq 1$$
> > > under which the bias bound of the DM estimator is strictly larger than that of the TPG estimator with truncation size $k$.
> > >
> > > For example, when $k=1$, this condition reduces to $\delta \leq 1$, which holds by definition. Therefore, the TPG estimator with $k=1$ always admits a strictly smaller bias bound than the DM estimator, regardless of the mixing coefficient $\gamma \in (0,1]$. When $k \geq 2$, the inequality here imposes a concrete regime for both $\delta$ and $\gamma$.
> > >
> > > **Further Intuition**
> > >
> > > For intuition, it is helpful to interpret the estimator through an *influence-function-type* perspective. The ATE can be viewed as the difference in long-run outcomes between the *fully treated* policy and the *fully controlled* policy, where the treatment assignment is applied over the entire time horizon. In this sense, the effect of a treatment at time $t$ propagates through the system and influences all future outcomes.
> > >
> > > In contrast, our approach introduces an alternative estimand that localizes this effect in time. Specifically, at each time $t$, we consider a counterfactual in which only the previous $k$ time steps are assigned to treatment (or control), while the remainder of the trajectory evolves under the baseline policy. This effectively truncates the temporal propagation of the treatment effect, so that we capture only its impact over a finite horizon of length $k$.
> > >
> > > From this perspective, the TPG estimator can be understood as estimating a *truncated influence function*: it retains the dominant short-term impact of the treatment while discarding long-range effects that are harder to estimate and contribute to bias.
> > >
> > > The role of mixing is precisely to control the discarded tail. Under a mixing condition, the influence of a treatment decays geometrically over time, which ensures that the contribution beyond horizon $k$ can be uniformly bounded by a term of order $\gamma^k/(1-\gamma)$. And there is a *magic trick* that comes from the Policy Gradient Theorem: after a reindexing, the backward-propagating effects can be equivalently expressed in terms of forward-looking quantities (i.e., the truncated $Q$-function). We refer the reviewer to our proof of Theorem~3.2 for a derivation of this correspondence.

---

### Official Review · Reviewer_h9XN · 2026-03-09

**Soundness:** 4
**Presentation:** 3
**Significance:** 3
**Originality:** 4
**Overall Recommendation:** 5
**Confidence:** 3

**Summary:**

The paper studies A/B testing in nonstationary environments. The goal is to estimate the global average treatment effect (GATE) between two policies. The author(s) develops a novel method by connecting the GATE with the policy gradient, a terminology used widely in policy-based reinforcement learning. The proposed estimator admits the form of the policy gradient, and is further truncated for MSE reduction. The author(s) established both the finite-sample properties of their estimators by investigating their biases and variances, and conducted numerical experiments to illustrate their usefulness.

**Compliance With Llm Reviewing Policy:**

Affirmed.

**Final Justification:**

As mentioned in the review, I appreciate the connection between A/B testing and policy gradient algorithms in RL. The proposed methodology is novel and statistically sound. I will keep my acceptance score.

**Key Questions For Authors:**

I am a bit curious about the differences between the theoretical results in Proposition 4.2 and Theorem 3.2. In Theorem 3.2, the second term in the bias bound grows exponentially with the horizon k, whereas in Proposition 4.2 the corresponding term grows linearly with the horizon T. The variance bounds also appear to differ. Additionally, some expressions include a gamma factor while others do not. Could the author(s) elaborate on the reasons for these differences?

2. Would it be possible to derive doubly robust version of the proposed estimator for further variance reduction? The proposed estimator appears more similar to the importance sampling estimator.

**Limitations:**

I might have missed something, but I did not find adequate discussions of the limitations.

**Strengths And Weaknesses:**

Strengths:

1. The paper studies A/B testing in nonstationary environments. While there is a huge literature on ATE estimation and A/B testing, less has been studied in nonstationary environments.

2. The proposed estimator is novel and is derived via a Taylor expansion. It also admits an interesting interpretation in terms of policy gradients. To the best of my knowledge, there has been little work explicitly bridging these two perspectives, and the paper does a very nice job highlighting this connection.

3. The proposed estimator is theoretically justified and its usefulness is demonstrated through numerical experiments, including both synthetic studies and simulations based on real-world environments.

Some (potential) weaknesses:

1. Page 2, Line 129: The citation “(Hu & Wager, 2022)” should be written in citet form as “Hu & Wager (2022)”.

2. Remark 3.1 is not entirely accurate. While direct estimators and robust estimators require state information, sequential importance sampling estimators with state-agnostic randomization probabilities do not.

3. It would be helpful to clarify the type of nonstationarity considered in the paper. In general, there are two common settings: (i) Episodic setting, where each episode has a finite horizon and a new episode is drawn afterward. In this case, the non-stationarity occurs only within an episode, while the episodes themselves follow identical distributions across repetitions (see, e.g., https://arxiv.org/pdf/2403.17285). (ii) Continuous setting, where only a single trajectory (episode) is observed and the transition dynamics evolve continuously over time.
The current paper appears to study the second setting.

4. The CLT in Theorem 6.3 is a nice result. However, based on the presented results, the inference appears to apply to the expected value of the estimator rather than the true target GATE. When the bias is large, this expectation may differ from the true target.

5. The numerical section is rather brief, spanning only one page, with many details relegated to the appendix. This somewhat hurts readability. However, this issue could be addressed by using the additional page in the camera-ready version if the paper is accepted.

---

> ### Author Rebuttal · Authors · 2026-03-31
>
> **We thank the reviewer for their careful reading and constructive comments.**
>
> Weaknesses:
> 1. We thank the reviewer for the careful reading. We will correct this citation format in the camera-ready version if accepted.
> 2. We agree that Remark 3.1, as originally stated, was overly broad. State-agnostic sequential importance sampling estimators can indeed avoid explicit state modeling (see, e.g., Xie et al. (2019)). Our intended point is more specific: TPG avoids both explicit state modeling and the use of long-horizon importance weights. It should not have been phrased as suggesting that all robust alternatives necessarily require state information. We will revise this remark to make the distinction precise in the camera-ready version.
> 3. Our paper studies a continuous single-trajectory setting, i.e., we observe a single finite-horizon path, the transition kernels $P_t$ evolve over time, and there are no episodic resets to a common initial distribution. Moreover, we do not impose structural restrictions or parametric assumptions on the form of this nonstationarity, except for the regularity conditions required to construct a consistent HAC variance estimator. This distinction is important, as our bias and variance analysis rely on within-trajectory randomized perturbations and temporal mixing, rather than averaging across repeated i.i.d. trajectories. A similar continuous-time nonstationary framework is studied in Hu and Wager (2022).
> 4. The reviewer is correct that the stated CLT is centered at $\mathbb{E}[\hat{\tau}_k]$. To obtain valid inference for the target $\tau$, it is additionally necessary that the bias $|\mathbb{E}[\hat{\tau}_k] - \tau|$ be negligible at the $1/\sqrt{T}$ scale. Theorem 3.2 provides an upper bound of order $O\left(k^2 \delta^2 M + \frac{\gamma^k \delta M}{1 - \gamma}\right)$; however, this bound is generally not negligible at the CLT scale, making it challenging to construct fully bias-robust confidence intervals for $\tau$. Accordingly, the objective of Section 6 is to establish consistent variance estimation for the TPG estimator under nonstationarity. We note that such CLTs are commonly employed when the bias is not asymptotically negligible (see, e.g., Farias et al. (2022)).
> 5. We thank the reviewer for the suggestion. We will expand the main text to provide a more comprehensive and self-contained presentation, using the additional page if the paper is accepted.
>
> Key Questions:
> 1. For the untruncated estimator, the bias is bounded by $O(T^2)$ in the worst case without mixing, while under a mixing condition, it becomes uniformly bounded but remains second-order relative to the $O(T^2)$ variance (as illustrated in Figure 2 and Table 2). Under mixing, truncation offers a clear advantage: it substantially reduces the variance while introducing only a small additional mixing bias (captured by the second term in both the existing Theorem 3.2 and the refined bound above). Furthermore, we note that the $O(T^2)$ behavior in Proposition 4.2 is intended to characterize the worst-case nonstationary regime, where no geometric decay can be exploited (i.e., $\gamma \to 1$).
> 2. We have considered this question and experimented with a doubly robust variant of the TPG estimator. The main challenge in applying doubly robust techniques is obtaining meaningful value estimates for state-action pairs in a nonstationary MDP. In our setting, the value function depends not only on the state but also explicitly on time, due to the nonstationarity. With only a single trajectory, estimating such time-varying values is not practical. Nevertheless, we conducted preliminary experiments using a naive value estimator that ignores time dependence and treats the value as a function of the state alone. The numerical results indicate that this approach does not lead to meaningful variance reduction, largely because the nonstationarity violates the stationarity assumptions required for effective doubly robust correction.
>
> Reference:
>
> Xie, T., Ma, Y., & Wang, Y. X. (2019). Towards optimal off-policy evaluation for reinforcement learning with marginalized importance sampling. Advances in neural information processing systems, 32.
>
> Hu, Y., & Wager, S. (2022). Switchback experiments under geometric mixing. arXiv preprint arXiv:2209.00197.

---

> > ### Author Rebuttal · Reviewer_h9XN · 2026-04-01
> >
> > Thank the authors for the efforts. My concerns have been addressed. I will keep my acceptance score.

---

### Official Review · Reviewer_wArF · 2026-03-13

**Soundness:** 3
**Presentation:** 2
**Significance:** 2
**Originality:** 3
**Overall Recommendation:** 4
**Confidence:** 3

**Summary:**

This paper proposed Truncated Policy Gradient (TPG) estimator, which replaces instantaneous outcomes with short-horizon outcome trajectories. The authors also established a central limit theorem for the proposed estimator and developed its variance estimator under non-stationarity with single-trajectory data. The proposed method is evaluated on two real-world case studies.

**Compliance With Llm Reviewing Policy:**

Affirmed.

**Final Justification:**

I recommend weak accept. The rebuttal has sufficiently addressed my concerns.

**Key Questions For Authors:**

1. Line 210: Can authors discuss more why the estimators defined in Eq(2) can mitigate temporal difference bias? It seems not straightforward solely from the formula.
2. Following up on my previous point, I was wondering why authors call this estimator “Truncated policy gradient”?  There doesn’t seem to be a gradient component in the estimator, and policy gradient normally refers to the policy optimization algorithm which may cause confusion.
3. Line 285: I think Proposition 4.2 require more discussion. How is it related to Proposition 4.1, how does the result from 4.1 extends to more general policies beyond uniform random policy?
4. For section 6, I was wondering will similar CLT hold if we have more than one trajectories, how will the convergence rate related to the number of trajectories?
5. Can author discuss if the current theoretical analysis can be extended to more general policy classes (e.g, if policy id dependent on current or previous state)? What would be the potential challenge?
6. For numerical experiments, I was wondering if authors can provide some analysis on the performance of the proposed estimator with various number of trajectories? Since most of the theoretical framework is developed under single trajectory, how do we deal with multiple trajectories empirically?

**Limitations:**

Yes.

**Strengths And Weaknesses:**

I think this paper is trying to tackle a very challenging problem in treatment effect identification under nonstationarity. The proposed estimator and theoretical analysis provides valuable insight for the problem, especially the idea of Taylor expansion approximation is very interesting, the overall analysis is also solid. However, the proposed framework is constrained on single trajectory and Markov policy, which limits the generalizability of the proposed framework. The proposed estimator is also quite sensitive to the selection of $k$ to achieve optimal performance, and is only evaluated on two simulated environments where tuning is relatively easy.

---

> ### Author Rebuttal · Authors · 2026-03-31
>
> **We thank the reviewer for their careful reading and constructive comments.**
>
> **Q1:**
> The key issue is that the traditional DM estimator uses only the immediate reward $Y_u$, and therefore ignores the effect of action $Z_u$ on future states and subsequent rewards $\{Y_{u+1}, \dots, Y_{u+k}\}$. This omission leads to temporal difference bias in dynamic settings, as it completely ignores temporal spillover effects.
>
> In contrast, the TPG estimator replaces $Y_u$ with the short-horizon return $\sum_{t=u}^{u+k} Y_t$, thereby incorporating the *first-order downstream effect* of the treatment assignment up to a truncation horizon $k$.
>
> In our paper, Proposition 5.1 exactly formalizes this property: $\mathbb{E}[\hat{\tau}_k]$ corresponds to a truncated policy gradient, i.e., the marginal effect of perturbing the treatment probability on the cumulative reward over the next $k$ periods. In this sense, the estimator accounts for the causal impact of the action on future rewards within the truncation window, which directly mitigates temporal difference bias.
>
> **Q2:**
> Proposition 5.1 (Eq. (13)) shows that $\mathbb{E}[\hat{\tau}_k] = \nabla J_k(1/2)$, i.e., it equals the derivative of the truncated policy value with respect to the Bernoulli treatment probability, evaluated at the experimental policy. This is why we refer to the estimator as a \emph{truncated policy gradient}: it corresponds exactly to the policy gradient of a (hypothetical) Bernoulli treatment policy (cf. Eq. (10)--(11)).
>
> We also clarify that the term is used in an *evaluative* sense: the estimator targets the gradient of a policy value functional associated with a hypothetical randomized policy, rather than serving as a component of a policy optimization or improvement procedure.
>
> **Q3:**
> Proposition 4.1 and Proposition 4.2 play complementary roles. Proposition 4.1 is an \emph{identity result}: it expresses the policy gradient as an average Q-value difference, thereby identifying the target of the PG estimator.
>
> Proposition 4.2 is an *approximation result*: since $\tau = J(1) - J(0)$, the gap between the policy gradient evaluated at $1/2$ and the full treatment–control difference can be characterized as a Taylor remainder. Thus, Proposition 4.2 quantifies how far the PG target may deviate from the true GATE, i.e., the bias decomposition.
>
> Moreover, the gradient identity in Proposition 4.1 is not specific to $\theta = 1/2$. As noted in Footnote 2, the same derivation extends directly to any Bernoulli policy $\pi_\theta$; in fact, it suffices to replace $1/2$ with a general $\theta$ throughout Proposition 4.1. We focus on $\theta = 1/2$ because it corresponds to the canonical A/B design and yields a particularly clean and symmetric form.
>
> **Q4:**
> The paper focuses on the more challenging single-trajectory setting. In contrast, when multiple trajectories are available, statistical inference becomes more straightforward due to the presence of i.i.d. samples; related settings have been studied in the OPE literature for (partially observed) stationary environments (see, e.g., Hu and Wager (2023)).
>
> If one has $N$ independent trajectories, a natural extension in our case is to compute $\hat{\tau}_k$ on each trajectory and average the resulting estimates. Since Theorem 6.3 establishes $\sqrt{T}$-scale fluctuations for a single trajectory (at fixed $k$), averaging $N$ independent estimates would heuristically reduce the variance by a factor of $1/N$, yielding a $\sqrt{NT}$ convergence rate.
>
> Hu, Y., & Wager, S. (2023). Off-policy evaluation in partially observed Markov decision processes under sequential ignorability. The Annals of Statistics, 51(4), 1561-1585.
>
> **Q5:**
> Extending the analysis to state- or history-dependent policies is possible in principle, but the current *state-free structure* would no longer hold. In particular, the score term would depend on the realized state or history, and the corresponding Q-value differences would become state-specific. Crucially, note that one practical advantage of our estimator is that it can be viewed as a plug-in replacement, applicable wherever the DM estimator can be used.
>
> In the single-trajectory, nonstationary setting considered in this paper, this introduces additional challenges, including state-coverage requirements and potential instability due to importance weighting (which may require augmenting the state with the time index).
>
> **Q6:**
> In the experiments, we simulate multiple trajectories primarily to assess sampling behavior (e.g., mean, variance). Specifically, the estimator is computed independently on each trajectory (i.e., applied in a trajectory-wise manner), and we then summarize the resulting estimates across trajectories (e.g., 500 trajectories in Case Study I and 100 i.i.d. trajectories in Case Study II).
>
> Empirically, if multiple i.i.d. trajectories are available, one can further reduce variability by directly averaging the TPG estimates obtained from multiple single trajectories.

---

> > ### Author Rebuttal · Reviewer_wArF · 2026-04-03
> >
> > Thank you authors for the detailed response, which sufficiently addressed my concerns. I think this is an interesting work, and I'll increase my rating to weak accept.

---

### Official Review · Reviewer_8XkB · 2026-03-13

**Soundness:** 3
**Presentation:** 3
**Significance:** 3
**Originality:** 2
**Overall Recommendation:** 4
**Confidence:** 3

**Summary:**

The paper addresses the challenging problem of estimating the global average treatment effect (GATE) in dynamic, nonstationary environments where interventions exhibit temporal carryover effects. The authors propose the Truncated Policy Gradient (TPG) estimator, which truncates the sequence of future rewards to balance bias and variance. Theoretical bounds for the bias and variance are provided under a uniform mixing time assumption, alongside a nonstationary Central Limit Theorem (CLT) and a Heteroskedasticity- and Autocorrelation-Consistent (HAC) variance estimator. The proposed method is evaluated empirically on a queueing simulator and a large-scale NYC ride-sharing simulator.

**Compliance With Llm Reviewing Policy:**

Affirmed.

**Final Justification:**

I have raised my score to Weak Accept because the authors' transparent and rigorous rebuttal successfully resolved my primary concerns. By revising Theorem 3.2 to accurately reflect that truncation provides significant variance reduction rather than rescuing an exploding bias under mixing conditions, the paper's theoretical claims are now sound. Additionally, their acknowledgment of the theory-practice gap regarding periodic data (Assumption 6.2) and the new robustness study for the heuristic parameter $\alpha$ alleviate my practical reservations. With these clarifications, the proposed estimator is a mathematically robust and practically valuable contribution to the field.

**Key Questions For Authors:**

## 1. Clarification on Assumption 6.2 and Periodic Data:
Assumption 6.2 (Cesaro-$L^2$ mean stability) requires the temporal deviation $\Delta_T^2$ to be bounded by $o(T^{-1/3})$. In Case Study II, the paper utilizes NYC ride-sharing data, which naturally exhibits strong periodic diurnal patterns (as visualized in Figure 6). Theoretically, a strictly periodic treatment effect and expectation sequence $\mu_t$ would result in a mean squared deviation $\Delta_T^2$ that converges to a positive constant ($\mathcal{O}(1)$), which seems to be at odds with the $o(T^{-1/3})$ requirement.

- Could the authors clarify how the nonstationary HAC estimator theoretically accommodates periodic nonstationarity?

- If Assumption 6.2 is primarily a regularizing assumption for theoretical tractability, it would be helpful to explicitly discuss this theory-practice gap and explain why the HAC estimator remains empirically robust in Case Study II despite the inherent periodicity.

## 2. Questions regarding the Bias Bound in Proposition 4.2 (Appendix E.2):
A core motivation for truncation is that the untruncated PG estimator suffers from an $\mathcal{O}(T^2)$ bias (Proposition 4.2). In the proof provided in Appendix E.2, terms A and B are bounded by $\mathcal{O}((u-1)\delta^2 M)$ and $\mathcal{O}((t-u-1)\delta^2 M)$ respectively, leading to the $\mathcal{O}(T^2)$ bound. However, for term C (Line 1189), the proof notes that under Assumption 2.1 (Mixing Time), the term enjoys exponential decay $\mathcal{O}(\gamma^{t-u-1})$.

- If Assumption 2.1 is consistently applied to the transition kernels within the summations of terms A and B, would the bounds instead yield a convergent geometric series $\sum \gamma^i \le \frac{1}{1-\gamma}$, leading to an $\mathcal{O}(1)$ overall bias bound for the untruncated estimator?

- The reviewer would be grateful if the authors could clarify the mathematical rationale for not applying the mixing property to terms A and B, and whether the $\mathcal{O}(T^2)$ explosion is strictly necessary if the environment is already assumed to mix uniformly.

## 3. Sensitivity to the Mixing Time Assumption (Assumption 2.1):
The theoretical guarantees depend heavily on the uniform mixing time property. In highly congested real-world scenarios (e.g., severe weather events in ride-sharing or massive sudden influxes in hospitals), a system might remain in a specific state for an extended period, potentially violating the fast mixing assumption ($\gamma \to 1$).

- Does the TPG estimator's performance degrade gracefully under these conditions? A brief sensitivity analysis or discussion on the behavior of the estimator when $\gamma$ approaches 1 would significantly strengthen the paper's practical claims.

## 4. Heuristic Selection of $k$ (Algorithm 1):
Algorithm 1 provides a practical method for selecting the truncation size $k$ based on a user-defined stability threshold $\alpha$.

- How sensitive are the final GATE estimates to the choice of $\alpha$? Since relying on a hyperparameter might introduce the risk of manual tuning in A/B testing, could the authors provide an ablation study or further guidelines on selecting $\alpha$ robustly?

**Limitations:**

The authors should explicitly discuss the theory-practice gap regarding periodic data. Real-world systems like ride-sharing naturally exhibit diurnal patterns, which theoretically result in a mean squared deviation $\Delta_T^2$ of $\mathcal{O}(1)$, violating the $o(T^{-1/3})$ requirement of Assumption 6.2. Acknowledging this gap or providing a relaxed bound for periodic variations is necessary. Furthermore, the paper lacks a discussion on how the TPG estimator's performance might degrade in highly congested scenarios where the uniform mixing time assumption ($\gamma$ approaching 1) is severely violated.

**Strengths And Weaknesses:**

## 1. Soundness

- Strengths: The empirical evaluation is rigorously designed, utilizing complex, realistic simulators (a nonstationary queueing system and a large-scale NYC taxi environment) rather than oversimplified toy models. The TPG estimator demonstrates clear empirical advantages over stationary baselines.

- Weaknesses:
     - Mathematical Inconsistencies: As mentioned above, the proof in Appendix E.2 inconsistently applies Assumption 2.1. Furthermore, the claim that periodic ride-sharing data strictly satisfies the Cesaro-$L^2$ mean stability (Assumption 6.2) is mathematically contradictory and needs to be addressed or relaxed.
     - Extreme Conditions: The paper lacks an ablation or sensitivity analysis demonstrating how the TPG estimator degrades if the uniform mixing assumption is severely violated (e.g., when $\gamma \to 1$ during extreme system congestion).
     - Heuristic Reliance: Algorithm 1 relies heavily on a user-defined stability threshold $\alpha$ to select the truncation size $k$. This introduces a heuristic step without rigorous theoretical guarantees, potentially allowing for hyperparameter sensitivity in practical A/B testing.



## 2. Presentation

- Strengths: The submission is generally well-written and logically structured. The narrative smoothly guides the reader from the failures of traditional A/B testing into the Markovian framework, making complex RL concepts accessible to a broader causal inference audience.

- Weaknesses: There are a few minor notational issues that hinder clarity. In Algorithm 1, the stopping criterion uses $\hat{\sigma}_k$, which is defined in the text as the asymptotic variance. Dimensional consistency suggests this should likely be the standard error (scaling with $1/\sqrt{T}$). Additionally, there is a swapping of dummy variables ($t$ and $u$) between Equation (2) and its counterpart in Appendix D.1, which should be harmonized for readability.


## 3. Significance
- Strengths: The paper tackles a highly relevant and challenging problem: estimating the global average treatment effect (GATE) in dynamic, nonstationary environments with temporal carryover effects. The proposed Truncated Policy Gradient (TPG) estimator offers strong practical utility for real-world platforms (e.g., digital health, ride-sharing) because it relies solely on observed rewards and does not require full state observability.

- Weaknesses: The practical significance of the proposed nonstationary HAC variance estimator is currently limited by a noticeable theory-practice gap. Real-world systems like NYC ride-sharing exhibit strong periodic diurnal patterns, which inherently result in a mean squared deviation $\Delta_T^2$ of $\mathcal{O}(1)$. This strictly contradicts the $o(T^{-1/3})$ bound required by Assumption 6.2, raising questions about the estimator's applicability to the very domains it targets.

## 4. Originality

- Strengths: Framing the difference of $k$-step accumulated rewards as a truncated policy gradient is a creative and insightful perspective. It successfully bridges reinforcement learning concepts with causal inference and experimental design, offering a fresh lens for evaluating dynamic treatments.

- Weaknesses: The theoretical novelty of the truncation mechanism appears somewhat overstated due to inconsistent bounding. The paper claims that truncation is theoretically necessary to rescue the baseline estimator from an $\mathcal{O}(T^2)$ bias explosion (Proposition 4.2). However, this explosion seems to be an artifact of selectively ignoring the uniform mixing property (Assumption 2.1) for terms A and B in the proof (Appendix E.2), while applying it to term C. If the mixing property is applied consistently, the untruncated bias would be bounded by $\mathcal{O}(1)$, which weakens the theoretical justification for the method's originality.

---

> ### Author Rebuttal · Authors · 2026-03-31
>
> **We thank the reviewer for the careful reading of our paper and appendix.**
>
> **Q1:**
> We note that Assumption 6.2 provides a *sufficient condition* for the CLT/HAC result; it is not intended to cover arbitrary fixed-amplitude periodic mean drift. As noted following Assumption 6.2, this form of mean stability is required to establish consistency from a single nonstationary trajectory. Accordingly, we do not claim that Assumption 6.2 holds for periodic processes with non-vanishing amplitude. Rather, Case Study II can be interpreted as demonstrating empirical robustness under approximate, real-world periodicity, which does not correspond to an exact fixed-amplitude periodic drift but instead exhibits a form of mean stability consistent. We find it particularly encouraging that, in such practical settings, the variance estimator remains well-calibrated and achieves accurate coverage. One possible direction to generalize the HAC variance estimator to periodic time series is to incorporate a seasonality-aware kernel that places more weight on autocovariances at seasonal lags (see, e.g., Shin and Oh, 2002).
>
> Shin, D. W., & Oh, M. S. (2002). A new kernel for long-run variance estimates in seasonal time series models. Economics Letters.
>
> **Q2:**
> **We sincerely thank the reviewer for this sharp observation.**
>
> As the reviewer noted, if under a mixing condition, one can apply the geometric decay within the summations in terms $A$ and $B$, yielding a double convergent series $\sum_j\sum_i \gamma^i\gamma^j \leq 1 /(1-\gamma)^2$. In this case, when $\theta = 1/2$ (as in Theorem 3.2), the bias of the untruncated estimator can be bounded by
> $
> O\left(\frac{\delta^2M }{(1-\gamma)^2}\right).
> $
>
> Based on this observation, we’ve refined Theorem 3.2 as follow:
>
> | Estimator | Bias | Variance |
> |-----------|------|----------|
> | TPG (finite $k$) | $O\left(\dfrac{\delta^2 M}{(1-\gamma)^2} + \dfrac{\gamma^k}{1-\gamma}\delta M\right)$ | $O\left(\dfrac{(k+1)^3 M^2}{T}\right)$ |
> | Untruncated PG ($k = T$) | $O\left(\dfrac{\delta^2 M}{(1-\gamma)^2}\right)$ | $O\left(T^2 M^2\right)$ |
>
> This table reveals the advantage of the TPG estimator: **it dramatically reduces the variance from $O(T^2)$ to $O(k^3/T)$ at the cost of a small truncation bias** (the second term, which decays exponentially in $k$).
>
> We also note that this refined bound directly reflects the reviewer’s comment on the behavior of the untruncated estimator under a mixing condition. In particular, if taking $k \to \infty $ (i.e., the untruncated case), the second term vanishes, and the bound recovers the mixing-based bound discussed above exactly. We are able to more precisely delineate the relationship between the untruncated and truncated estimator: For the untruncated estimator, the bias is bounded by $O(T^2)$ in the worst case without mixing, while under a mixing condition, it becomes uniformly bounded but remains second-order relative to the  $O(T^2)$ variance. Under mixing, truncation offers a clear advantage: it substantially reduces the variance while introducing only a small mixing bias. In addition, the $O(T^2)$ behavior in Proposition 4.2 is intended to characterize the worst-case nonstationary regime, where no geometric decay can be exploited ($\gamma \to 1$).
>
> In addition, this revision is lightweight: it requires only a minor refined analysis to the $\delta^2$ term in the bias bound and an update to the presentation of the tradeoff, without altering any other results or the structure of the paper.
> We thank the reviewer again for raising this point, which helps significantly clarify and strengthen our paper. We will incorporate these updates if the paper is accepted.
>
> **Q3:**
> We note that Appendix G.1 already provides a sensitivity analysis. Figure 4 illustrates the behavior as $\gamma$ increases: the MAE rises as $\gamma \to 1$, and the optimal truncation level shifts upward, though it remains small relative to $T=5000$. These empirical patterns are consistent with Theorem 3.2: slower mixing weakens the geometric decay of the tail term, making larger $k$ preferable, while still reflecting the inherent bias--variance trade-off.
>
> **Q4:**
> Our heuristic post-$k$ selection algorithm is in the spirit of the Lepski method. In practice, Appendix B adopts $\alpha = 1.96$ as a default choice, corresponding to a standard 95\% confidence level. In Case Study I, the resulting selection $k=5$ achieves near-nominal 95\% confidence interval coverage (Table~1). Thus, $\alpha$ primarily governs the stopping point along the bias--variance trade-off. In practice, it can be chosen according to standard normal quantiles (e.g., $[1.645, 1.96, 2.326, 2.576]$ corresponding to [90\%, 95\%, 98\%, 99\%] confidence levels).
>
> Following is a robustness study (we will include this study in the supplement):
>
> | Confidence | alpha | Median k | RMSE (vs ATE) |
> |---|---:|---:|---:|
> | 90% | 1.645 | 4 | 0.040 |
> | 95% | 1.960 | 3 | 0.049 |
> | 98% | 2.326 | 3 | 0.052 |
> | 99% | 2.576 | 3 | 0.054 |

---

> > ### Author Rebuttal · Reviewer_8XkB · 2026-04-03
> >
> > Thank the authors for their detailed and transparent rebuttal. As my theoretical and empirical concerns have been fully resolved, I will slightly raise my overall score accordingly.

---

### Decision · Program_Chairs · 2026-04-30

**Decision:**

Accept (regular)

**Comment:**

This paper studies treatment-effect estimation under nonstationarity and proposes the Truncated Policy Gradient (TPG) estimator. I find the contribution both technically interesting and well executed. The manuscript proposes new estimator and derives theory in a setting that is both practically relevant and still under-explored.

The review discussion was constructive and, in my reading, moved substantially in the paper's favor. Several technical concerns were explicitly resolved after rebuttal, and the remaining limitations are not fatal. Overall, this paper clears the acceptance bar comfortably. I therefore recommend Accept.